# Coverage-based Example Selection for In-Context Learning

**Shivanshu Gupta**[1]    **Matt Gardner**[2]    **Sameer Singh**[1]

[1]University of California Irvine    [2]Scaled Cognition

{shivag5,sameer}@uci.edu, mgardner@scaledcognition.com

## Abstract

In-context learning (ICL), the ability of large language models to perform novel tasks by conditioning on a prompt with a few task examples, requires these examples to be informative about the test instance. The standard approach of independently ranking and selecting the most similar examples selects redundant examples while omitting important information. In this work, we show that BERTScore-Recall (BSR) selects better examples that demonstrate more of the *salient aspects*, e.g. reasoning patterns, of the test input. We further extend BSR and many standard metrics to easily optimizable set-level metrics, giving still better coverage of those salient aspects. On 15 datasets spanning 6 tasks and with 7 diverse LLMs, we show that (1) BSR is the superior metric for in-context example selection across the board, and (2) for compositional tasks, set selection using Set-BSR outperforms independent ranking by up to 17 points on average and, despite being training-free, surpasses methods that leverage task or LLM-specific training.[1]

## 1 Introduction

Large language models (LLMs) (Devlin et al., 2019; Brown et al., 2020) are capable of generalizing to novel tasks (Brown et al., 2020) by conditioning on textual prompts consisting of a few task examples. This training-free paradigm of few-shot inference, known as in-context learning (ICL), reduces the cost of modeling new tasks while also providing an interpretable and customizable interface (Liu et al., 2022; Wei et al., 2023) and improving generalization (Anil et al., 2022; Qiu et al., 2022b; Drozdov et al., 2023) and reasoning skills (Wei et al., 2023). However, ICL performance is critically sensitive to the choice of demonstrations (Zhao et al., 2021; Liu et al., 2022; Lu et al., 2022; Rubin et al., 2022; Schick and Schütze,

---

[1]https://github.com/Shivanshu-Gupta/icl-coverage

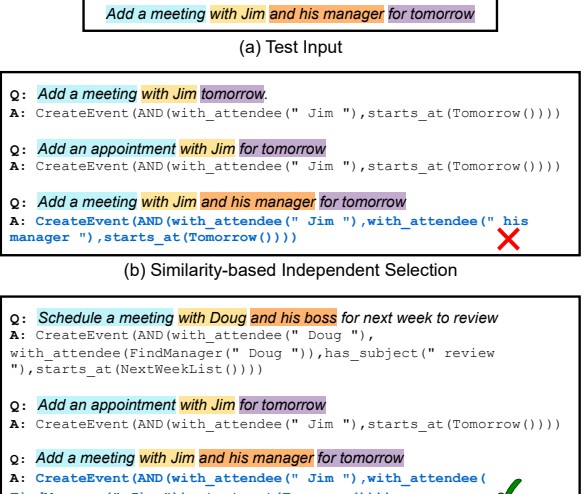

Figure 1: (a) Test input with salient aspects highlighted. (a) Independently selecting similar examples leads to redundancy and failure to demonstrate all salient aspects, in this case, the need to identify the manager. (b) Coverage-based selection using SET-BSR mitigates this by selecting a *less* similar example that contains the missing information. **Blue** indicates LLM generation.

2021), as the LLM relies on them for understanding and solving the test instance.

The standard approach to selecting ICL examples or demonstrations from a pool of candidates is to independently score them using a relevance metric and choose the top-ranked ones. However, cosine similarity and BM25, the two commonly used metrics, are sub-optimal for selecting demonstrations due to their reliance on a single dense embedding and unigram overlap, respectively. Moreover, since it selects examples independently, this approach ignores their utility as a set. It is particularly inadequate for complex compositional tasks like semantic parsing (Levy et al., 2022) where no single candidate might contain all reasoning patterns, and an independent selection would select multiple redundant examples with the same reasoning patterns but fail to demonstrate the others. Figure 1 shows a

failure case where similarity-based selection picks paraphrased examples that fail to demonstrate how to find a manager. Prior work on selecting demonstrations as a set (Ye et al., 2023; Levy et al., 2022) required task and/or LLM-specific training, limiting their utility. For this reason, simple yet widely applicable training-free methods like BM25 and cosine similarity remain the most popular approaches for ICL example selection.

In this work, we propose a novel framework for selecting sets of maximally informative demonstrations for the salient aspects of the test input, e.g., reasoning patterns, entities, etc. Examples selected using this framework are informative about the test input and help the LLM understand and perform the task. We use this framework to explore different ways to characterize salient aspects, including syntactic structures like dependency parse subtrees and contextual token embeddings, while using BM25 and BERTScore (Zhang et al., 2020) to measure their coverage, respectively. To select the demonstrations as a set, we extend the coverage metrics to measure the overall informativeness of a set of demonstrations. We show that these set-level metrics are submodular and can be efficiently optimized to find demonstration sets that maximally cover the salient aspects.

We evaluate our ICL example selection methods on 15 diverse datasets, including 6 semantic parsing, 2 numerical reasoning, and 7 classification datasets, and with 7 LLMs of varying sizes and pretraining. Among instance-level metrics, BSR, the recall version of BERTScore, consistently outperforms standard retrieval metrics on all datasets and LLMs, beating cosine similarity by up to 8 points on average in semantic parsing datasets and 15 points in the rest. Selecting demonstrations as a set using SET-BSR, the set-extension of BSR, leads to further gains in semantic parsing and is particularly effective in compositional settings where the gains grow with LLM size. With Codex, a 175B parameter LLM, SET-BSR outperforms cosine similarity by 17% on average with up to 49% improvement in some splits, and, despite being training-free, outperforms even trained methods like those from Rubin et al. (2022), Levy et al. (2022), and Ye et al. (2023) that require task and/or LLM-specific training.

## 2   Related Work

In-context learning for few-shot inference facilitates the use of LLMs for novel tasks without the need for expensive supervised fine-tuning. In addition to reduced cost, it has several other advantages over supervised fine-tuning: it provides a more interpretable and customizable interface to using LLMs (Liu et al., 2022; Wei et al., 2023); and retention of linguistic understanding and knowledge from pretraining leading to improved generalization (Anil et al., 2022; Qiu et al., 2022b; Drozdov et al., 2023) and reasoning skills (Wei et al., 2023).

However, the performance of ICL is critically sensitive to the choice of demonstrations (Zhao et al., 2021; Liu et al., 2022). This has led to a growing interest in techniques for selecting good demonstrations. Prior work can be roughly classified into (1) independently scoring and retrieving examples (Liu et al., 2022; Rubin et al., 2022), (2) selecting diverse examples to reduce redundancy among them (Su et al., 2022; Levy et al., 2022; Agrawal et al., 2022; Ye et al., 2022), and (3) selecting examples that minimize the entropy of the LLM's output distribution for the test input (Lu et al., 2022; Wu et al., 2023). Recent work has also trained RL agents (Lu et al., 2023) and used Bayesian inference (Wang et al., 2023).

The most similar studies to ours are Levy et al. (2022) and Ye et al. (2023). Levy et al. (2022) select diverse demonstrations that cover substructures of the target output predicted by task-specific classifiers but are limited in applicability to a few semantic parsing tasks. Ye et al. (2023) use Determinantal Point Processes (Kulesza, 2012) to select a diverse set of demonstrations similar to the test instance but do not optimize for coverage directly and require training with the LLM. Moreover, both methods require task or LLM-specific training that limits their use and effectiveness for larger LMs.

## 3   Preliminaries

**In-context learning** is the ability of LLMs to solve novel tasks by merely conditioning on a few task demonstrations. Formally, given demonstrations $\{(x_i, y_i)\}_{i=1}^{k}$ and the test input $x_{\text{test}}$, it involves using textual templates to linearize instance inputs and outputs into sequences of tokens from the LLM vocabulary, $\mathbf{x} = \mathcal{I}(x) = \langle x_1 \ldots x_{|\mathbf{x}|} \rangle$ and $\mathbf{y} = \mathcal{O}(y) = \langle y_1 \ldots y_{|\mathbf{y}|} \rangle$. The linearizations are then concatenated to form a prompt and fed to the LLM for conditional generation of the test output:

$$\mathbf{y}_{\text{test}} \sim \mathcal{P}_{LM} \left( \cdot \mid \mathbf{x}_1, \mathbf{y}_1, \ldots, \mathbf{x}_K, \mathbf{y}_K, \mathbf{x}_{\text{test}} \right) \quad (1)$$

The interpretable and training-free nature of ICL makes it an attractive alternative to supervised fine-tuning. However, its performance is highly sensitive to the choice and order of demonstrations.

**Demonstration Selection** identifies which examples to include in the prompt for any test instance. Formally, given a test input $x_{\text{test}}$ and a pool of candidates $\mathcal{T} = \{z_i\}_{i=1}^N = \{(x_i, y_i)\}_{i=1}^N$, the goal is to select a subset of $k \ll N$ demonstrations that when included in the context make $\mathbf{y}_{\text{test}}$ the most likely generation. A naive approach is to randomly sample $k$ instances from $\mathcal{T}$, but this is sub-optimal since the demonstrations are often completely unrelated to the test input. Instead, the standard approach to selecting demonstrations that are informative about the test input is to independently assign each candidate $z$ a score $\texttt{score}(x_{\text{test}}, z)$ using a relevance metric and then select the top $k$ candidates.

**Relevance Metrics** The two most commonly used relevance metrics for scoring demonstration are cosine similarity and BM25. Cosine similarity uses a representation function $\mathcal{R}$ to independently map the textual linearizations of inputs to unit-norm embeddings $\mathbf{r}_x = \mathcal{R}(x)$ in a common vector space and then scores the candidate $z$ using the dot product, $\texttt{cosine}(x_{\text{test}}, z) = \mathbf{r}_{x_{\text{test}}}^T \mathbf{r}_z$. BM25, on the other hand, is a sparse information retrieval algorithm belonging to a class of TF-IDF measures that view the test input and the candidates as bags of terms and measures relevance as a weighted recall or coverage of these terms:

$$\texttt{tfidf}(x_{\text{test}}, z) = \sum_{s \in T_{x_{\text{test}}}} \texttt{idf}(s)\texttt{tf}(s, T_z) \quad (2)$$

Here $T_x$ and $T_z$ are the set of terms in $x$ and $z$ respectively, and $\texttt{tf}(s, T_z)$ and $\texttt{idf}(s)$ are the term frequency and inverse document frequency statistics that measure the coverage of a particular term and the relative importance of terms respectively. We use $\texttt{tf}$ and $\texttt{idf}$ as per the Okapi variant of BM25 (Robertson et al., 1993; Jones et al., 2000).

## 4 Informative Demonstrations

The limitation of the standard demonstration selection approach is that by independently scoring the demonstrations, it ignores their utility as a set. For ICL to work, the demonstrations included in the context need to be informative about how to understand and solve the test input. In this section

and the next, we describe our approach to selecting informative sets of demonstrations for ICL. We begin by defining our notion of informativeness of demonstrations in ICL and describing how to measure it. Thereafter, in §5, we will discuss how to extend this notion to an algorithm for selecting optimally informative sets of demonstrations.

**Informativeness** Demonstrations should demonstrate the salient aspects, e.g., reasoning patterns, entities, etc., of the test input. Formally, denoting $S_{x_{\text{test}}}$ as the set of salient aspects of the test input, we measure the informativeness of a demonstration $z$ in terms of the coverage of such salient aspects,

$$\texttt{cover}(x_{\text{test}}, z) = \sum_{s \in S_{x_{\text{test}}}} \texttt{c}(s, z) \quad (3)$$

where $\texttt{c}(s, z)$ measures the coverage (or recall) of a single salient aspect $s$ by $z$.

**Salient Aspects** Both cosine similarity and BM25 are special cases of Eq. 3 for different notions of salient aspects. For BM25, $S_{x_{\text{test}}} = T_{x_{\text{test}}}$, the set of unigrams in $x$, and $\texttt{c}(s, z) = \texttt{idf}(s)\texttt{tf}(s, T_z)$. And cosine similarity, although not explicitly a recall metric, can also be interpreted as evaluating coverage of the dimensions of the test input embedding by defining $S_{x_{\text{test}}} = [1, d]$, the dimensions of the dense embedding as the salient aspects, i.e.,

$$\texttt{cosine}(x_{\text{test}}, z) = \sum_{s=1}^d \mathbf{r}_{x_{\text{test}}}[s] \cdot \mathbf{r}_z[s] \quad (4)$$

The above interpretations reveal why neither cosine similarity nor BM25 are good measures of informativeness. While cosine similarity captures some aspects of semantic similarity (depending on the embedding), it is limited to a single embedding. And, unigrams, the commonly used terms with BM25, are too small to capture most salient aspects. A good measure of informativeness necessitates an accurate characterization of salient aspects. One way might be to use larger syntactic substructures of the input as terms with BM25. We experiment with using larger n-grams and subtrees of the dependency parse tree. However, such syntactic structures are constrained to the surface form of the instance and hence may not capture meaning and aspects like reasoning patterns. A better way to capture salient aspects is to use contextualized token embeddings, the idea behind the BERTScore (Zhang et al., 2020) metric.

**BERTScore** was originally proposed as a metric for evaluating the quality of machine-generated text (e.g., machine translation) by comparing it to a reference text. It leverages pre-trained contextual embeddings to match words in the candidate and reference sentences by cosine similarity and compute precision, recall, and F1 measures. Formally, given the sequences of contextual embeddings $\langle \mathbf{x}_1, \mathbf{x}_2, \ldots, \mathbf{x}_{|x|}\rangle$ and $\langle \mathbf{z}_1, \mathbf{z}_2, \ldots, \mathbf{z}_{|z|}\rangle$ of tokens in $x = \langle x_1, x_2, \ldots, x_{|x|}\rangle$ and $z = \langle z_1, z_2, \ldots, z_{|z|}\rangle$ respectively, the recall measure, BERTScore-Recall (BSR), is defined as:

$$\text{BSR}(x, z) = \sum_{x_i \in x} w(x_i) \max_j \mathbf{x}_i^T \mathbf{z}_j \qquad (5)$$

Here, $w(x_i)$ is a weight assigned to token $x_i$ and can be defined as $\frac{1}{|x|}$ if treating each token as equally important or $\frac{\text{idf}(x_i)}{\sum_{x_i \in x} \text{idf}(x_i)}$ if downweighting rare words. The precision measure is defined analogously, while the F1 measure is the harmonic mean of the two. BSR is also a special case of Eq. 3 with contextualized tokens as salient aspects, i.e., $S_x = \langle \mathbf{x}_1, \mathbf{x}_2, \ldots, \mathbf{x}_{|x|}\rangle$ and can be used to select examples by treating them as candidates and the test input as the reference. The following table summarizes the informativeness measures and salient aspects in this work.

| Metric | Salient Aspects |
|---|---|
| Cosine | embedding dimensions |
| BM25 | unigrams, n-grams, dependency parse subtrees |
| BERTScore | contextual token embeddings |

## 5 Set-level Information Coverage

So far, we have focused on measuring the informativeness of a single demonstration to rank and independently select the most informative ones. However, as depicted in Fig. 1, when no single single candidate demonstrates all salient aspects, this approach can fail to cover all of them while also selecting redundant demonstrations that provide no new information. A scenario where this can happen is when the candidate pool contains close paraphrases (or duplicates). This suggests that demonstrations should be selected as a set.

**Set Metric** To evaluate the informativeness of a set of examples $Z$, we propose to extend the coverage measure in Eq. 3 to a measure for sets as follows:

$$\text{setcov}\left(x_{\text{test}}, Z\right) = \sum_{s \in S_{x_{\text{test}}}} \max_{z \in Z} \mathsf{c}(s, z) \qquad (6)$$

---

**Algorithm 1** Greedy Optimization of Set Coverage

**Require:** Instance pool $\mathcal{T}$; test input $x_{\text{test}}$; desired number of demonstrations $k$; coverage scoring function `setcov`
1: $Z \leftarrow \emptyset$ ▷ Selected Demonstrations
2: $Z_{\text{curr}} \leftarrow \emptyset$ ▷ Current Set Cover
3: `curr_cov` $\leftarrow -\inf$
4: **while** $|Z| < k$ **do**
5: $\quad z^*, \texttt{next\_cov} = \underset{z \in \mathcal{T} - Z}{\arg\max} \, \texttt{setcov}\left(x_{\text{test}}, Z_{\text{curr}} \cup z\right)$
6: $\quad$ **if** `next_cov > curr_cov` **then** ▷ Pick $z^*$
7: $\qquad$ `curr_cov` $\leftarrow$ `next_cov`
8: $\qquad Z \leftarrow Z \cup z^*$
9: $\qquad Z_{\text{curr}} \leftarrow Z_{\text{curr}} \cup z^*$
10: $\quad$ **else** ▷ Or start new cover
11: $\qquad Z_{\text{curr}} \leftarrow \emptyset, \texttt{curr\_cov} \leftarrow -\inf$
12: $\quad$ **end if**
13: **end while**
14: **return** $Z$

---

Intuitively, this measures the coverage of each salient aspect as the *best* coverage it receives from *any* example in the set. In other words, maximizing it requires that every salient aspect appears at least once in *some* demonstration without considering which or how many. Since cosine similarity, BM25, and BSR are all special cases of Eq. 3, they can be extended to set measures using Eq. 6.

**Submodularity** Given the combinatorial space of sets of demonstrations, for a measure on sets to be practical, it needs to be efficiently optimizable. Fortunately, the set-level metric, as defined above, is also submodular for any definition of $\mathsf{c}(s, z)$. We prove this in Appendix A. Intuitively, this follows from the facts that (1) for any given test instance, $\mathsf{c}(s, z)$ assigns a scalar weight to each demonstration $z \in Z$, (2) the maximum of weights across set elements is submodular, and (3) the sum of submodular functions is also submodular. This means that the set-level metric can be optimized using a greedy algorithm with a constant factor approximation guarantee (Nemhauser et al., 1978).

**Algorithm** The greedy algorithm we use to select the optimal set is shown in Algorithm 1. In every iteration, it selects the example that maximally increases the coverage of the current set of demonstrations (lines 5-9). If no such example exists, it resets (lines 11). Using the following identity when computing the score for candidate sets (line 5),

$$\begin{aligned} \texttt{setcov}&\left(x_{\text{test}}, Z \cup z'\right) \\ &= \sum_{s \in S_{x_{\text{test}}}} \max\left(\mathsf{c}(s, Z), \mathsf{c}(s, z')\right) \end{aligned} \qquad (7)$$

and assuming constant time for computing each $\mathsf{c}(s, z)$, the time complexity of algorithm is

$\mathcal{O}(kNL)$, where $L = |S_{x_{\text{test}}}|$. For BSR, the complexity of computing $c(x, z)$ for all $z \in Z$ is $\mathcal{O}(Td)$, where $T$ is the total number of tokens in $Z$ and $d$ is the token embedding size. Thus, the time complexity of both instance and set-level BSR is dominated by the computation of $c(x, z)$, and is $\mathcal{O}(LTd)$. While slower than cosine and BM25, we found it to be a small overhead to in-context learning for most datasets considered in this work. We discuss this further in App. C.

# 6 Experimental Setup

## 6.1 Datasets

We experiment with a total of 15 datasets including six diverse semantic parsing datasets viz. **Geo-Query** (Zelle and Mooney, 1996), **ATIS** (Hemphill et al., 1990; Dahl et al., 1994), **Overnight** (Wang et al., 2015), **SMCalFlow** (Andreas et al., 2020), **BREAK** (Wolfson et al., 2020), and **MTOP** (Li et al., 2021); a math-word problems (**GSM8K** (Cobbe et al., 2021)) and a machine reading comprehension (**DROP** (Dua et al., 2019)) dataset requiring multi-step numeric reasoning; and seven classification datasets spanning natural language inference, paraphrase detection and sentiment classification viz. **QNLI** (Wang et al., 2018), **MNLI** (Williams et al., 2018), **RTE** (Bentivogli et al., 2009), **MRPC** (Dolan and Brockett, 2005), **PAWS** (Zhang et al., 2019), **QQP** (Wang et al., 2018), and **SST2** (Socher et al., 2013). We refer the reader to App. B for detailed descriptions of each dataset along with sample instances and prompt templates.

In addition to the standard IID splits, we also evaluate compositional generalization using compositional splits wherever available. For GeoQuery we use three types of compositional splits: Template (Finegan-Dollak et al., 2018), TMCD (Keysers et al., 2020), and Length. Following Levy et al. (2022), we use the compositional splits—three Template, three TMCD, and one Length—generated by Qiu et al. (2022a) and average results across the TMCD and Template splits. For ATIS and Overnight, we experiment with Template splits (Finegan-Dollak et al., 2018) generated by Gupta et al. (2022). For SMCalFlow, we experiment with splits in **SMCalFlow-CS** (Yin et al., 2021): an IID split (8-S) and a compositional split (32-C).

For all the splits, following prior work (Ye et al., 2023; Rubin et al., 2022) we randomly subsample 44,000 instances from the train set to use as pool to select demonstrations from. For evaluation, we use a random subsample of 1000 instance of the validation set if available, and the test set otherwise. We use Exact Match (EM) accuracy for all datasets except BREAK where we use LF-EM (Hasson and Berant, 2021), which is preferred over EM for semantic equivalence.

## 6.2 Models

We experiment with the following LLMs: **GPT-Neo-2.7B** (Black et al., 2021): A 2.7B-parameter LM trained on The Pile (Gao et al., 2020), an 825 GB text corpus. **LLaMA** (Touvron et al., 2023): A collection of LMs ranging from 7B to 65B parameters pretrained on CommonCrawl, GitHub, Arxiv, etc. We experiment with LLaMA-7B and LLaMA-13B. **StarCoder** (Li et al., 2023): A 15.5B parameter model trained on 80+ programming languages (Kocetkov et al., 2022). **GPT-3.5-Turbo**[2]: 175B LM trained with RL to follow instructions and optimized for chat. **Cushman, Codex**[3] (Chen et al., 2021): 12B and 175B parameter code-pretrained LMs. GPT-Neo-2.7B, LLaMA-7B, LLaMA-13B, and Cushman have context window lengths of 2048, GPT-3.5-Turbo of 4096, Codex of 8001, and StarCoder of 8192.

## 6.3 Methods

### 6.3.1 Training-Free Methods

We compare the following training-free metrics:

**Cosine similarity** (COSINE) We use the Sentence-Bert library (Reimers and Gurevych, 2019) with the `all-mpnet-base-v2` model. For independent selection, we use FAISS [4] (Johnson et al., 2019) retrieve the most similar examples.

**BM25** (BM25) We use the Okapi variant (Robertson et al., 1993; Jones et al., 2000) of BM25 from the `rank_bm25`[5] library with three syntactic structures as terms: unigrams, size-4 or smaller n-grams, and size-4 or smaller subtrees of the input dependency parse (obtained using the `spaCy`[6]).

**BERTScore** We use the `bert_score`[7] library (Zhang et al., 2020) with `deberta-large-mnli` and `deberta-base-mnli` models which are De-BERTa models (He et al., 2021) finetuned on the MNLI dataset (Williams et al., 2018). We will refer

---

[2] https://openai.com/blog/chatgpt/. We use the gpt-3.5-turbo-0301 snapshot from March 2023.

[3] We use code-davinci-002 and code-cushman-001.

[4] https://github.com/facebookresearch/faiss

[5] https://github.com/dorianbrown/rank_bm25

[6] https://spacy.io

[7] https://github.com/Tiiiger/bert_score

| | Selector | GPT-Neo | LLaMA-7B | LLaMA-13B | Cushman | StarCoder | GPT-3.5-Turbo | Codex |
|---|---|---|---|---|---|---|---|---|
| Training Free | RANDOM | 5.5 (-23.3) | 5.7 (-28.7) | 9.8 (-28.9) | 12.0 (-32.9) | 13.6 (-33.5) | 13.0 (-31.9) | 20.7 (-32.6) |
| | COSINE | 28.8 | 34.4 | 38.7 | 44.9 | 47.1 | 44.9 | 53.4 |
| | BM25 | 31.2 (+2.4) | 36.7 (+2.3) | 42.8 (+4.0) | 49.7 (+4.8) | 52.9 (+5.7) | 50.3 (+5.4) | 60.9 (+7.5) |
| Trained | EPR | **38.3** (+9.5) | 43.7 (+9.3) | 48.1 (+9.4) | 51.8 (+7.0) | 53.5 (+6.4) | 47.4 (+2.5) | 58.5 (+5.1) |
| | CEIL | 38.1 (+9.3) | **44.5** (+10.1) | 49.9 (+11.2) | 54.8 (+9.9) | 57.3 (+10.2) | 51.2 (+6.3) | 64.0 (+10.7) |
| Ours | BSR | 34.1 (+5.3) | 40.1 (+5.8) | 46.5 (+7.8) | 52.6 (+7.7) | 54.8 (+7.7) | 52.7 (+7.8) | 61.2 (+7.9) |
| | SET-BSR | 35.8 (+7.0) | 43.8 (+9.4) | **51.4** (+12.7) | **59.5** (+14.6) | **61.6** (+14.5) | **60.1** (+15.2) | **70.3** (+16.9) |

Table 1: Average 8-shot ICL performance across all splits of semantic parsing datasets using different LLMs and demonstration-selection methods with absolute improvement over COSINE in brackets. Both BSR and SET-BSR outperform prior training-free methods, with the latter outperforming even trained methods with larger LLMs.

to the recall, precision, and F1 variants as BSR, BSP, and BSF1, respectively. Unless specified otherwise, we do not apply importance weighting (IDF) and use `deberta-large-mnli`.

Additionally, we experiment with (1) a random baseline (RANDOM) that randomly selects demonstrations from the pool, and (2) with the set-extensions of COSINE, BM25 and BSR as described in §5 which will be referred to as **SET-COSINE**, **SET-BM25**, and **SET-BSR** respectively.

### 6.3.2 Trained Methods

We also compare with methods that require task or LLM-specific training. **EPR** (Rubin et al., 2022) uses LLM perplexity to train a dense retriever for each dataset. **CEIL** (Ye et al., 2023) uses EPR and an LLM to train a Determinantal Point Process (Kulesza, 2012) for each dataset and then uses it to select examples. We use Ye et al. (2023)'s implementation of EPR and CEIL and use GPT-Neo-2.7B LLM. We also compare with **LFCOV** (Levy et al., 2022), a method for semantic parsing, specifically SMCalFlow-CS and GeoQuery. It trains a classifier to predict logical form substructures and then selects diverse examples containing them. We use the shots provided by the authors.

### 6.4 Prompt Construction

For $k$-shot (we use $k = 8$ unless specified otherwise) ICL with any given dataset (§ 6.1), demonstration selection method (§ 6.3) and LLM (§ 6.2), we construct the prompt as follows: (1) select up to $k$ demonstrations depending on the context window of the LLM; (2) order the demonstrations in increasing order of relevance so that the most relevant demonstrations appear closest to the test input; and (3) linearize the ordered demonstrations and the test input using the dataset's prompt template in Table 5 and concatenate to form the prompt. For set-selection methods, the demonstrations are or-

| | Selector | 8_S | 32_C |
|---|---|---|---|
| Training Free | RANDOM | 31.9 (-22.8) | 7.4 (-4.5) |
| | COSINE | 54.7 | 11.9 |
| | BM25 | 65.4 (+10.7) | 29.4 (+17.5) |
| Trained | EPR | 76.3 (+21.6) | 21.7 (+9.8) |
| | CEIL | **77.5** (+22.8) | 40.1 (+28.2) |
| | LFCOV | 66.3 (+11.6) | 45.9 (+33.9) |
| Ours | BSR | 72.5 (+17.8) | 31.5 (+19.6) |
| | SET-BSR | 75.7 (+21.0) | **61.2** (+49.3) |

Table 2: 8-shot ICL accuracy on SMCalFlow-CS using Codex with absolute improvement over COSINE in brackets. SET-BSR is competitive with trained methods on the IID split while dramatically outperforming them on the compositional split.

dered by their corresponding instance-level score. For the trained baselines, we use orderings recommended by the corresponding authors.

## 7 Results

We begin by comparing the performance of our proposed methods, BSR and SET-BSR, with prior training-free and state-of-the-art trained methods in § 7.1. We then analyze the different metrics for measuring informativeness of individual demonstrations (§ 7.2) and the impact of coverage-based set selection using our set extension (§ 7.3).

### 7.1 Main Results

Table 1 compares average performance across all semantic parsing splits for seven LLMs of varying sizes. See Table 2 for comparison with LFCOV, which only works with GeoQuery and SMCalFlow-CS and Table 11 for results on individual splits. While BSR consistently outperforms COSINE and BM25 for all LLMs, set-selection using SET-BSR leads to further dramatic gains with upto 17% improvement over COSINE with Codex, beating even state-of-the-art trained methods like EPR and CEIL by 12 and 6 points, respectively. Further, from

| | Selector | GSM8K | DROP | MNLI | PAWS | SST2 |
|---|---|---|---|---|---|---|
| Training Free | **Random** | 60.6 | 62.7 | 41.9 | 48 | 86.9 |
| | **Cosine** | 64 | 65.4 | 44.0 | 52.5 | 81.9 |
| | **BM25** | 64.8 | 66.9 | 42.2 | 55.2 | 82.6 |
| Trained | **EPR** | 61.7 | - | 66.1† | - | - |
| | **CEIL** | 63.1 | - | 71.7† | - | - |
| Ours | **BSR** | **68.1** | **68.1** | 76.7 | 75 | 90.9 |
| | **Set-BSR** | 67.4 | 66.4 | 78.6 | 74.9 | 61.5 |

Table 3: 8-shot ICL performance for tasks other than semantic parsing (using GPT-Neo-2.7B for the classification tasks and Codex for the harder GSM8K and DROP). BSR is competitive with prior methods, however, as these are IID splits, SET-BSR doesn't lead to further gains. † 50-shot results from Ye et al. (2023).

| Selector | ALL | IID | COMP |
|---|---|---|---|
| **BSF1** | 60.6 | 71.0 | 50.1 |
| **BSP** | 54.3 | 65.5 | 43.2 |
| **BSR** | 61.2 | 71.5 | 50.9 |
| **BM25** | 60.9 | 68.9 | 52.8 |
| + Coverage | 56.4 | 63.4 | 49.5 |
| **BM25[4-gram]** | 59.1 | 67.1 | 51.0 |
| + Coverage | 64.5 | 68.9 | 60.2 |
| **BM25[4-depst]** | 57.8 | 65.5 | 50.0 |
| + Coverage | 64.9 | 68.6 | 61.2 |

Table 4: Average 8-shot ICL performance with Codex on **IID**, **COMP**ositional, and **ALL** semantic parsing splits. Top compares different variants of BERTScore, white Bottom compares the different variants of BM25.

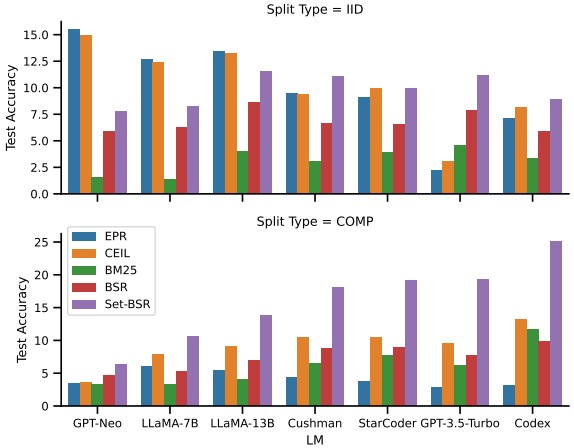

Figure 2: Gain in average ICL accuracy compared to COSINE on **IID** and **COMP**ositional splits in semantic parsing. Trained methods (EPR and CEIL) become less effective with larger LLMs on IID splits. This is unlike SET-BSR, which, on compositional splits, even becomes more effective with larger LLMs.

Table 3, we see that, unlike SET-BSR, BSR is effective even for non-semantic parsing datasets outperforming COSINE by 15 points on average with GPT-Neo-2.7B (see Table 12), and often even EPR and CEIL (see Table 13). All the above improvements were statistically significant ($p < 0.05$) under paired permutation-tests.

**SET-BSR is more effective with larger LLMs** The effectiveness of SET-BSR monotonically improves as LLMs become more powerful. The trend is particularly pronounced in compositional splits, where it gets 25% absolute improvement v/s COSINE on average (see Fig. 2) and 49% improvement on the 32-C split of SMCalFlow-CS (see Table 2).

**Trained methods do not leverage larger LLMs** As EPR and CEIL are trained using GPT-Neo-2.7B, they have difficulty generalizing to and taking ad-

vantage of larger, more powerful LLMs, becoming less effective on IID splits (Fig. 2), and failing on GSM8K (Table 3). The latter is likely because GPT-Neo-2.7B itself fails on GSM8K (Table 13), which requires Chain-of-Thought reasoning, an emergent ability of larger LLMs (Wei et al., 2022). As training with increasingly large LLMs is prohibitively expensive and impractical, these results demonstrate serious limitations of trained methods.

### 7.2 Measure of Informativeness

**Contextual embeddings capture salient aspects** From Tables 1 and 3, it is clear that BSR consistently outperforms COSINE and BM25. This is true even when using the same encoder (see App. D), is seen in both IID and compositional splits (see Fig. 2), and with varying number of demonstrations (see Fig. 4). Larger syntactic substructures did not improve BM25 as seen in Table 4 (Bottom). These results show that contextual embeddings are indeed better at capturing salient aspects.

**Recall outperforms other measures** Comparing the variants of BERTScore, for Codex in Table 4 (Top), and other LLMs in Fig. 7 in App. D, it is evident that recall is on par with, or better than, the F1 metric. This supports our hypothesis that recall or coverage (of salient aspects) is a useful metric for informativeness. We include additional ablations in App. D, analyzing the effect of using importance weighting (IDF) and using a larger LM to compute token embeddings for BSR.

### 7.3 Coverage-based Set Selection

**Impact on performance** From Fig. 3, we see that coverage-based set selection is most effective in compositional splits where it improves the average performance of all metrics, including COSINE.

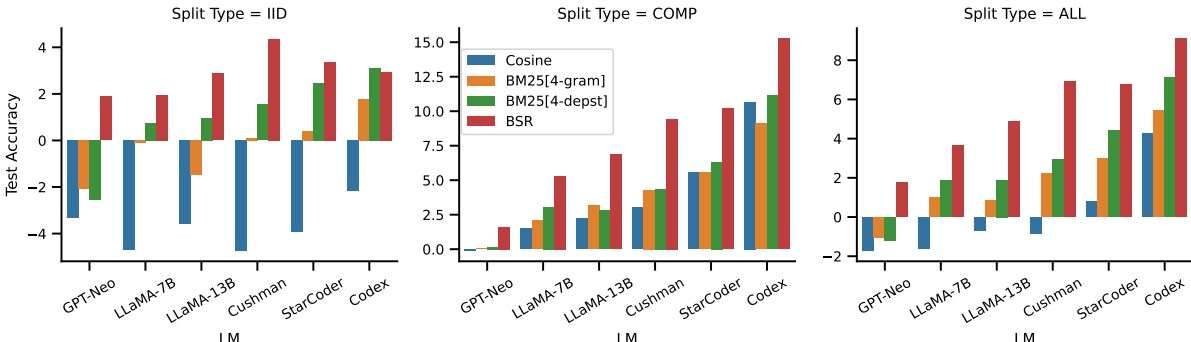

Figure 3: Change in average performance on different types of splits of semantic parsing datasets from set-selection using our set metrics v/s the corresponding instance-level metric. Coverage-based set selection is most useful in compositional splits and when covering larger syntactic structures (BM25) or contextual embeddings (BSR).

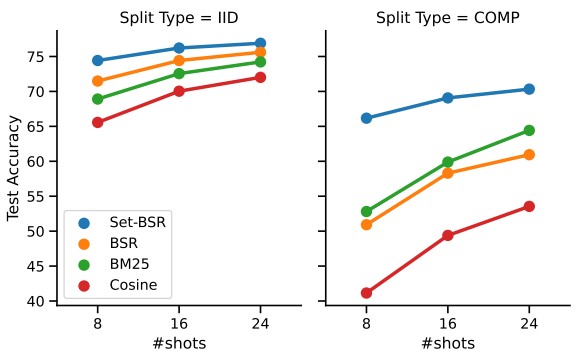

Figure 4: Average performance on **IID** and **COMP** semantic parsing splits with Codex. SET-BSR consistently outperforms independent selection.

This shows the importance of selecting demonstrations as a set in compositional settings where examples demonstrating all the salient aspects of the test input are even less likely to exist. The set extension is less effective in IID splits and even hurts performance for COSINE and vanilla unigram BM25. Overall, BSR and BM25 with larger substructures benefit the most from the set extension. We provided further analyses of improvements from set selection and the impact of reordering in App. D.

**Illustrative Example** We present a GeoQuery test input in Fig 5 along with demonstrations (only the inputs) selected by COSINE and SET-BSR (more examples in Appendix E). COSINE selections tend to be redundant, with repeated operations, and are somewhat restrictive, mostly limited to the `min` operation. Contrastingly, SET-BSR exhibits a more balanced selection, opting for demonstrations of comparable complexity to the test instance and collectively encapsulating all necessary operations.

**Failure Cases** There are a few limitations of coverage-based set-selection using SET-BSR. First, by only considering uncovered aspects, it sacrifices

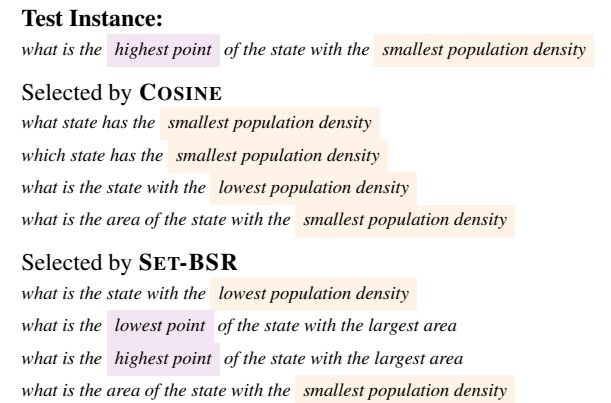

Figure 5: Demonstrations selected for a GeoQuery input (outputs omitted for clarity). COSINE demonstrations are redundant (repeated operations) and limited (only cover "population" aspect). SET-BSR, instead, selects demonstrations that are similarly complex as the test instance and, together, cover all required operations.

the relevance of individual demonstrations to prioritize coverage of all aspects with the set (see Table 9 for an example from GSM8K). Additionally, even contextual token embeddings can only capture salient aspects that are explicitly expressed in the input text and thus may not be suitable for tasks where the salient aspects are more abstract and require reasoning themselves (see Table 10 for an example from QNLI). We leave it to future work to explore better measures of informativeness, including better characterizations of salient aspects.

## 8 Conclusion

This paper presents a novel framework for selecting informative sets of demonstrations that cover salient aspects of the test input to aid the language model (LLM) in solving it. We explore different ways to characterize these aspects and quantify their coverage. Evaluation on a wide

range of tasks and LLMs validates the effectiveness of BERTScore-Recall as a measure of informativeness of individual demonstrations. Further, our results demonstrate the superiority of SET-BSR in selecting informative sets of demonstrations compositional tasks like semantic parsing and highlight the ability of coverage-based demonstration selection, unlike trained methods, to leverage increasingly powerful larger LLMs. Our code base is available at `https://github.com/Shivanshu-Gupta/icl-coverage`.

## Acknowledgements

We would like to thank the anonymous reviewers for their feedback. This work was sponsored in part by the DARPA MCS program under Contract No. N660011924033 with the United States Office Of Naval Research and in part by the NSF award #IIS-2046873. The views expressed are those of the authors and do not reflect the policy of the funding agencies.

## Limitations

Contextual token embeddings require the salient aspects to be expressed in text and hence may not be able to capture them for all tasks. Moreover, since it requires computing a dot product for every pair of test and candidate instance tokens, this causes it to scale quadratically with the average number of tokens making it computationally infeasible for tasks with very long textual linearizations. Future work can thus explore more general characterizations of salient aspects and more efficient methods for selecting demonstrations covering them.

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

## A Submodularity

**Definition A.1** (Submodular Function). If $\Omega$ is a finite set, a submodular function is a set function $f : 2^\Omega \to \mathbb{R}$, where $2^\Omega$ denotes the power set of $\Omega$, which satisfies one of the following equivalent conditions.

1. For every $X, Y \subseteq \Omega$ with $X \subseteq Y$ and every $x \in \Omega \backslash Y$ we have that $f(X \cup \{x\}) - f(X) \geq f(Y \cup \{x\}) - f(Y)$.

2. For every $S, T \subseteq \Omega$ we have that $f(S) + f(T) \geq f(S \cup T) + f(S \cap T)$.

3. For every $X \subseteq \Omega$ and $x_1, x_2 \in \Omega \backslash X$ such that $x_1 \neq x_2$ we have that $f(X \cup \{x_1\}) + f(X \cup \{x_2\}) \geq f(X \cup \{x_1, x_2\}) + f(X)$.

**Theorem A.1.** *The function $f_{maxw}(X) = \max\limits_{x \in X} w_x$ is submodular for any assignment of weights $w_x$ to the elements $x \in \Omega$.*

*Proof.* The following are clearly true for any $x \in \Omega$ and any $x_1, x_2 \in \Omega$ such that $w_{x_1} > w_{x_2}$:

1. $f_{\mathrm{maxw}}(X \cup \{x\}) \geq f(X)$

2. $f_{\mathrm{maxw}}(X \cup \{x_1\}) = f_{\mathrm{maxw}}(X \cup \{x_1, x_2\})$

Adding these two inequalities together, we get the third definition of submodularity and thus $f_{\mathrm{maxw}}$ is submodular. $\qquad\square$

**Theorem A.2.** *If $\{f_i\}_{i=1}^n$ are all submodular functions, then $\sum\limits_{i=1}^n f_i$ is also submodular.*

*Proof.* We show this for $n = 2$:

$$
\begin{aligned}
(f_1 + f_2)&(X_1 \cup X_2) + (f_1 + f_2)(X_1 \cap X_2) \\
&= (f_1(X_1 \cup X_2) + f_1(X_1 \cap X_2)) \\
&\quad + (f_2(X_1 \cup X_2) + f_2(X_1 \cap X_2)) \\
&\leq (f_1(X_1) + f_1(X_2)) + (f_2(X_1) + f_2(X_2)) \\
&= (f_1 + f_2)(X_1) + (f_1 + f_2)(X_2)
\end{aligned}
$$

$$(8)$$

Therefore, $f_1 + f_2$ is submodular using the second definition of submodularity. By induction, this is true for any number $n$ of functions. $\qquad\square$

**Theorem A.3.** *The set-level coverage metric $\mathtt{setcov}(x_{test}, Z)$ as defined in Eq. 6 is submodular for any definition of $\mathtt{c}(s, z)$.*

*Proof.* From Theorem A.1, the function $f_s(Z)$ defined as $f_s(Z) = \max\limits_{z \in Z} \mathtt{c}(s, z)$ is submodular for any definition of $\mathtt{c}(s, z)$. Further, since from Theorem A.2, the sum of submodular functions is also submodular, $\mathtt{setcov}(x_{\text{test}}, Z) = \sum\limits_{s \in S_{x_{\text{test}}}} f_s(Z)$ is submodular. $\qquad\square$

## B Datasets

We use 15 diverse datasets, including 6 semantic parsing, 2 numerical reasoning, and 7 classification datasets.

### B.1 Semantic Parsing

We use 6 semantic parsing datasets with IID and compositional splits for our experiments. Table 5 shows sample instances from each dataset we experiment with along with the textual template we use to linearize the instances. The ICL prompt is constructed by concatenating the templatized demonstrations and the test instance using \n\n as the separator.

**GeoQuery** (Zelle and Mooney, 1996): A dataset containing 880 natural language questions about US geography paired with Prolog programs. In addition to the standard (IID) split, we experiment with three types of compositional splits: (1) Template split where the training and test sets have disjoint program templates (Finegan-Dollak et al., 2018); (2) TMCD split which creates train and test sets with maximal compound divergence and minimal atom divergence (Keysers et al., 2020); and (3) Length split which evaluates for length generalization by testing on sequences longer than ones in training. Following Levy et al. (2022), we use the compositional splits — three Template, three TMCD, and one Length — generated by Qiu et al. (2022a) and average results across the TMCD and Template splits.

**ATIS** (Hemphill et al., 1990; Dahl et al., 1994): A dataset of natural language queries about aviation paired with $\lambda$-calculus programs. We experiment with an IID split and a Template split (Finegan-Dollak et al., 2018) for evaluating compositional generalization, both taken from (Gupta et al., 2022).

**Overnight** (Wang et al., 2015): A dataset containing both synthetic and natural language utterances from 11 domains (e.g. *socialnetwork, restaurants,* etc.) paired with Lambda-DCS logical forms. We experiment with an IID and a Template split of

the *socialnetwork* domain taken from (Gupta et al., 2022).

**SMCalFlow** (Andreas et al., 2020): A dataset of task-oriented natural language dialogs about calendars, weather, places, and people paired with executable dataflow programs. **SMCalFlow-CS** (Yin et al., 2021) is a subset of SMCalFlow containing single-turn dialogs involving two domains (organization structure and calendar event creation), each having its own set of program symbols with two types of test sets: a cross-domain (C) test set containing only instances where both domains appear and meant to test for compositional generalization, and a single-domain (S) test set contains instances with only single-domain for in-distribution evaluation. For compositional evaluation, we use the 32-C split which is a few-shot cross-domain split where the training set includes 32 cross-domain examples. For our IID evaluation, following Levy et al. (2022), we use the 8-S split. Additionally, we use the programs with the simplified syntax provided by (Meron, 2022).

**BREAK** (Wolfson et al., 2020) is a dataset that maps complex natural language questions into a language-based meaning representation (QDMR) comprising an ordered list of atomic steps necessary to answer the question. Following (Rubin et al., 2022), we use the low-level Break subset where the targets are logical forms comprising lists of operators with their arguments based on the corresponding QDMR.

**MTOP** (Li et al., 2021): A multilingual task-oriented semantic parsing dataset spanning six languages and 11 domains. The target commands are complex queries featuring nested intent-slot prediction. We use the English subset of MTOP from (Rubin et al., 2022).

### B.2 Non-Semantic Parsing

We additionally experiment with the standard IID splits of 9 non-semantic parsing datasets from the following categories:

**Numerical Reasoning**: For this category, we experiment with **GSM8K** (Cobbe et al., 2021), a chain-of-thought reasoning (Wei et al., 2023) dataset of grade school-level arithmetic reasoning problems expressed in natural language and **DROP** (Dua et al., 2019), a dataset of question-answer pairs where the questions are about paragraphs containing numerical information and the answers are spans in the paragraph.

**Classification**: For this category, we experiment with three Natural Language Inference (NLI) datasests (**QNLI** (Wang et al., 2018), **MNLI** (Williams et al., 2018), and **RTE** (Bentivogli et al., 2009)), three Paraphrase Detection datasets (**MRPC** (Dolan and Brockett, 2005), **PAWS** (Zhang et al., 2019), and **QQP** (Wang et al., 2018)) and one Sentiment Classification dataset (**SST2** (Socher et al., 2013)).

## C  Selection Time

Despite their $\mathcal{O}(LTd)$ time complexity, we found example selection using both BSR and SET-BSR to be fast enough to not be a bottleneck to in-context learning for most datasets considered in this work. By using a GPU to compute $c(x, z)$s, we could get both to work in the order tens of milliseconds per test input on average which was significantly faster than the LLM inference time itself. The exceptions were DROP, PAWS, QQP, MNLI and QNLI for which the selection took >1 second due to much longer instances and/or larger instance pool. We leave it to future work to explore more efficient ways to measure informativeness.

## D  Additional Analyses

**BM25** From Fig. 6 we can see that coverage-based selection using BM25 with larger substructures outperforms vanilla unigram BM25 in compositional splits.

**BERTScore-Recall** Examining the impact of importance weighting in Fig. 8 which compares the performance change with using importance weighting (IDF) in BSR, we can see that its effect is not consistent across different LLMs. We also did not see any consistent improvement from using larger deberta-large-mnli for computing token embeddings for instance-level BSR (see Fig. 9). However, it did help with set-level selection using SET-BSR.

**Reordering** We found the reordering of demonstrations according to the corresponding instance-level metric to only be necessary for smaller LLMs (see Fig. 10), with it even hurting the performance of larger LLMs. We believe this is because larger and code-pretrained LLMs are more capable at composing the salient aspects in the different demonstrations and taking advantage of the full context.

**BSR outperforms Cosine even with the same encoder** In § 7.2, we showed that BSR with

| Dataset | Example Template | Sample Instance |
|---|---|---|
| Overnight | {source}\t{target} | **source:** *employees who finish after alices birthday*
**target:** `(call listValue (call getProperty ((lambda s (call filter (var s) (call ensureNumericProperty (string employment_end_date)) (string >) (call ensureNumericEntity (call getProperty en.person.alice (string birthdate)))))) (call domain (string employee))) (string employee)))` |
| ATIS | {source}\t{target} | **source:** *give me the flights from pittsburgh to los angeles thursday evening*
**target:** `( lambda $0 e ( and ( flight $0 ) ( during_day $0 evening : pd ) ( from $0 pittsburgh : ci ) ( to $0 los_angeles : ci ) ( day $0 thursday : da ) ) )` |
| GeoQuery | {source}\t{target} | **source:** *which river traverses most states*
**target:** `answer ( most ( river, traverse_2, state ) )` |
| SMCalFlow | {source}\t{target} | **source:** *Please put a 2 o'clock on my schedule where I'm meeting with boss Daniel.*
**target:** `CreateEvent(AND(with_attendee(" Daniel "),starts_at(NextTime(time=NumberPM(2)))))` |
| BREAK | {source}\t{target} | **source:** *Is there another cube that is the same size as the cyan cube; what color is it?*
**target:** `return the cyan cube ;return size of #1 ;return cubes besides #1 ;return sizes of #3 ;return #3 where #4 is the same as #2 ;return color of #5` |
| MTOP | {source}\t{target} | **source:** *latest news from washington times please*
**target:** `[IN:GET_STORIES_NEWS [SL:DATE_TIME latest ] [SL:NEWS_TYPE news ] [SL:NEWS_SOURCE washington times ] ]` |

Table 5: Semantic Parsing Datasets with corresponding sample instances and example templates used in for ICL.

`deberta-large-mnli` outperforms Cosine with `all-mpnet-base-v2`. Tables 15, 16, 17, and 18 show that the same trend holds even when using the same encoder, `bert-base-uncased`, for both metrics confirming that contextual embeddings are indeed better at capturing salient aspects.

**Recall of Syntactic Structures** The improvements from set-based selection may be explained by Fig. 11 where we see that set-extensions COSINE and unigram BM25 reduce the recall of substructures of the test input whereas the recalls increase with set-extensions of both BM25[4-GRAM] and BM25[4-DEPST], and even BSR, which does not explicity consider these substructures.

## E Qualitative Analysis of Prompts

Tables 7, 8 show demonstrations selected using COSINE and SET-BSR for instances from MTOP and SMCalFlow-CS respectively. In each case, CO-SINE find demonstrations that are all very similar to the test input but fails to demonstrate some salient aspect, whereas BSR selects less similar instances but ensures complete coverage of all salient aspects. Tables 9 and 10 additionally illustrate limitations of set-selection and of token-embeddings in capturing salient aspects.

## F All Results

Tables 11 contains 8-shot ICL results for our proposed methods and prior learning-free and learning-based demonstration selection on all the LLMs for all the semantic parsing datasets. For numerical reasoning and classification datasets, Tables 12 and 13 compare 8-shot ICL performance with prior training-free and trained methods, respectively. Table 14 provides average performances across all datasets.

Additionally, Tables 15, 16, 17, 18, 20, and 21 contain results on semantic parsing datasets of all ablations of learning-free selection methods we ran, with GPT-Neo-2.7B, LLaMA-7B, LLaMA-13B, StarCoder, Cushman, and Codex, respectively. We did not run ablations on GPT-3.5-Turbo due to its cost.

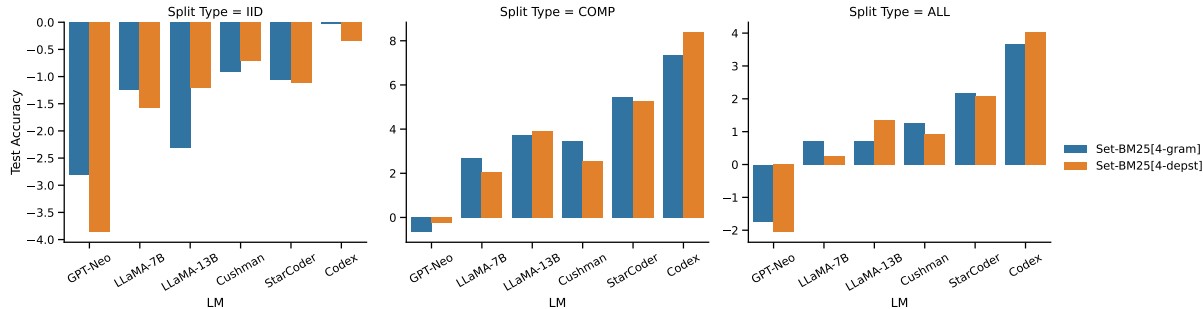

Figure 6: Absolute improvement in average 8-shot ICL performance on different types of semantic parsing splits from using the set extensions SET-BM25 with larger substructures over vanilla BM25.

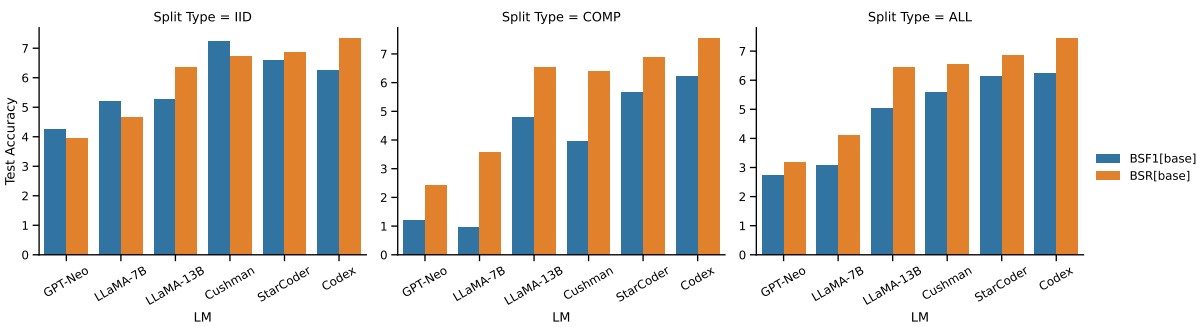

Figure 7: Comparison of 8-shot ICL performance of different variants of BERTScore with token embeddings computed using `deberta-base-mnli`. For easier visualization, since we found BERTScore-Precision to consistently perform worst, we show absolute improvement in average performance on different types of splits from the recall and F1 metrics over the precision metric.

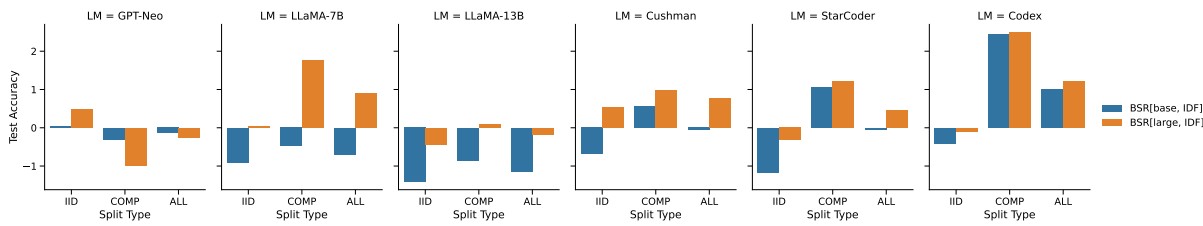

Figure 8: Impact on average 8-shot ICL performance on semantic parsing splits from using importance weighting (IDF) in BSR.

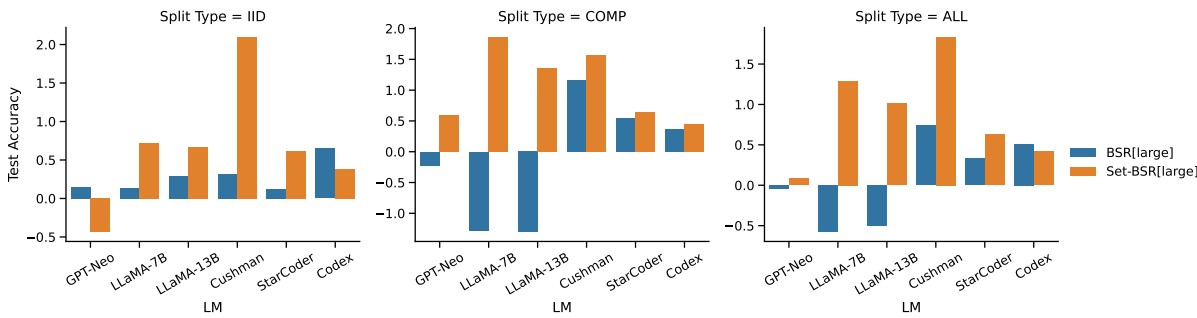

Figure 9: Impact on average 8-shot ICL performance on semantic parsing splits from using a larger `deberta-large-mnli` LLM for computing contextual token embeddings v/s using `deberta-base-mnli` in BSR and SET-BSR.

| Dataset | Example Template | Sample Instance |
|---|---|---|
| GSM8K | Question: {question}
Solution: {solution} | **question:** *Natalia sold clips to 48 of her friends in April, and then she sold half as many clips in May. How many clips did Natalia sell altogether in April and May?*
**solution:** *Natalia sold 48/2 = «48/2=24»24 clips in May. Natalia sold 48+24 = «48+24=72»72 clips altogether in April and May. #### 72* |
| DROP | Passage: {passage}
Question: {question}
Answer: {answer} | **passage:** *To start the season, the Lions traveled south to Tampa, Florida to take on the Tampa Bay Buccaneers. The Lions scored first in the first quarter with a 23-yard field goal by Jason Hanson. The Buccaneers tied it up with a 38-yard field goal by Connor Barth, then took the lead when Aqib Talib intercepted a pass from Matthew Stafford and ran it in 28 yards. The Lions responded with a 28-yard field goal. In the second quarter, Detroit took the lead with a 36-yard touchdown catch by Calvin Johnson, and later added more points when Tony Scheffler caught an 11-yard TD pass. Tampa Bay responded with a 31-yard field goal just before halftime. The second half was relatively quiet, with each team only scoring one touchdown. First, Detroit's Calvin Johnson caught a 1-yard pass in the third quarter. The game's final points came when Mike Williams of Tampa Bay caught a 5-yard pass. The Lions won their regular season opener for the first time since 2007*
**question:** *How many points did the buccaneers need to tie in the first?*
**answer:** *3* |
| QNLI | Question: {question}
Sentence: {sentence}
Answer: {label} | **sentence:** *Unlike the two seasons before it and most of the seasons that followed, Digimon Tamers takes a darker and more realistic approach to its story featuring Digimon who do not reincarnate after their deaths and more complex character development in the original Japanese.*
**question:** *When did the third Digimon series begin?*
**label:** *No* |
| MNLI | Premise: {premise}
Hypothesis: {hypothesis}
Answer: {label} | **premise**: *The new rights are nice enough*
**hypothesis**: *Everyone really likes the newest benefits*
**label**: *Maybe* |
| RTE | Premise: {premise}
Hypothesis: {hypothesis}
Answer: {label} | **premise**: *Dana Reeve, the widow of the actor Christopher Reeve, has died of lung cancer at age 44, according to the Christopher Reeve Foundation.*
**hypothesis**: *Christopher Reeve had an accident.*
**label**: *Yes* |
| MRPC | Sentence 1: {sentence1}
Sentence 2: {sentence2}
Answer: {label} | **sentence1**: *He said the foodservice pie business doesn't fit the company's long-term growth strategy.*
**sentence2**: *" The foodservice pie business does not fit our long-term growth strategy .*
**label**: *Yes* |
| PAWS | Sentence 1: {sentence1}
Sentence 2: {sentence2}
Answer: {label} | **sentence1**: *Bradd Crellin represented BARLA Cumbria on a tour of Australia with 6 other players representing Britain , also on a tour of Australia .*
**sentence2**: *"Bradd Crellin also represented BARLA Great Britain on a tour through Australia on a tour through Australia with 6 other players representing Cumbria .*
**label**: *No* |
| QQP | Question 1: {question1}
Question 2: {question2}
Answer: {label} | **question1**: *Why are African-Americans so beautiful?*
**question2**: *"Why are hispanics so beautiful?*
**label**: *No* |
| SST2 | Review: {sentence}
Answer: {label} | **sentence**: *it 's a charming and often affecting journey .*
**label**: *Positive* |

Table 6: Non-Semantic Parsing Datasets with corresponding sample instances and example templates used for ICL.

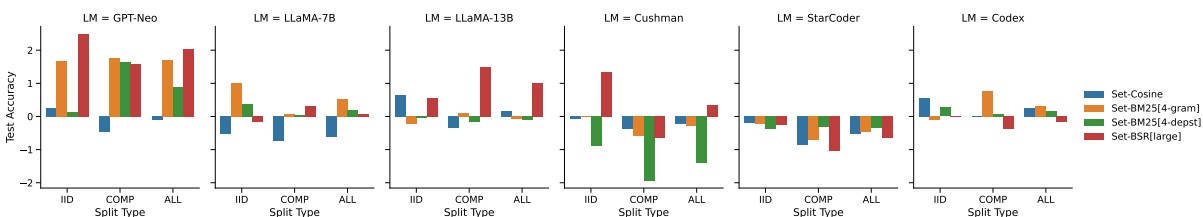

Figure 10: Impact on average 8-shot ICL performance on semantic parsing splits from reordering the demonstrations selected by the different set-level metric using the corresponding instance-level metric as absolute gain v/s the unreordered version.

| Selector | Prompt |
|---|---|
| COSINE | Sentence: Easy vegan recipes
Logical Form: [IN:GET_RECIPES [SL:RECIPES_ATTRIBUTE Easy ] [SL:RECIPES_TYPE vegan ] ]

Sentence: Vegetarian recipes
Logical Form: [IN:GET_RECIPES [SL:RECIPES_TYPE Vegetarian ] ]

Sentence: Please find me vegan recipes
Logical Form: [IN:GET_RECIPES [SL:RECIPES_TYPE vegan ] ]

Sentence: Give me vegan recipes
Logical Form: [IN:GET_RECIPES [SL:RECIPES_TYPE vegan ] ] |
| SET-BSR | Sentence: I have a nut allergy. Find me a dessert recipe
Logical Form: [IN:GET_RECIPES [SL:RECIPES_EXCLUDED_INGREDIENT nut ] [SL:RECIPES_MEAL dessert ] ]

Sentence: Create a video message for Victoria with plan options for dinner with family next week
Logical Form: [IN:SEND_MESSAGE [SL:TYPE_CONTENT video ] [SL:RECIPIENT Victoria ] ]

Sentence: What are some no-bake dessert ideas
Logical Form: [IN:GET_RECIPES [SL:RECIPES_COOKING_METHOD no - bake ] [SL:RECIPES_MEAL dessert ] ]

Sentence: Vegan birthday cakes
Logical Form: [IN:GET_RECIPES [SL:RECIPES_TYPE Vegan ] [SL:RECIPES_DISH birthday cakes ] ] |

Table 7: Demonstrations selected for the MTOP input: Vegan desert options with target output `[IN:GET_RECIPES [SL:RECIPES_TYPE Vegan ] [SL:RECIPES_DISH birthday cakes ] ]`. COSINE's reliance on a single dense embedding means it is unable to account for the fact that "options" could mean dishes and not just recipes.

| Selector | Prompt |
|---|---|
| COSINE | Sentence: I need a meeting with Elli tomorrow at 11 pm
Logical Form: CreateEvent(AND(with_attendee(" Elli "),starts_at(Tomorrow()),starts_at(NumberPM(11))))

Sentence: Set a meeting with Elli for tomorrow at 2 pm through the end of the day and call it Recap
Logical Form: CreateEvent(AND(ends_at(AND(GE(DateTime?(date=Tomorrow(),time=NumberPM(2))),EndOfWorkDay())),with_attendee(" Elli "),has_subject(" Recap "),starts_at(Tomorrow()),starts_at(NumberPM(2))))

Sentence: Schedule a meeting with Elli for tomorrow at 4 pm through the end of the workday
Logical Form: CreateEvent(AND(ends_at(AND(GE(DateTime?(date=Tomorrow(),time=NumberPM(4))),EndOfWorkDay())),with_attendee(" Elli "),starts_at(Tomorrow()),starts_at(NumberPM(4))))

Sentence: Schedule a meeting with Elli from 4 PM until the end of the day tomorrow .
Logical Form: CreateEvent(AND(ends_at(AND(GE(DateTime?(date=Tomorrow(),time=NumberPM(4))),EndOfWorkDay())),with_attendee(" Elli "),starts_at(Tomorrow()),starts_at(NumberPM(4)))) |
| SET-BSR | Sentence: I need a doctor 's appointment on Wednesday morning .
Logical Form: CreateEvent(AND(has_subject(" doctor's appointment "),starts_at(Morning()),starts_at(NextDOW(" WEDNESDAY "))))

Sentence: I need to see Alice and her boss next Monday at 3 pm .
Logical Form: CreateEvent(AND(with_attendee(" Alice "),with_attendee(FindManager(" Alice ")),starts_at(NextDOW(" MONDAY ")),starts_at(NumberPM(3))))

Sentence: Schedule a meeting with Jake , Elli , and Jesse for Friday at 2 pm .
Logical Form: CreateEvent(AND(with_attendee(" Jesse "),with_attendee(" Jake "),with_attendee(" Elli "),starts_at(NextDOW(" FRIDAY ")),starts_at(NumberPM(2))))

Sentence: I need to schedule a meeting with Jeff 's supervisor Lynne for tomorrow at 10 AM .
Logical Form: CreateEvent(AND(with_attendee(" Lynne "),starts_at(Tomorrow()),starts_at(NumberAM(10)))) |

Table 8: Demonstrations selected for the SMCalFlow-CS input: Schedule a meeting with Elli and her manager 's boss tomorrow morning. SET-BSR is able to find demonstrations covering all fragments of the test input while COSINE fails to include anything which involves finding someones manager.

| Selector | Prompt |
|---|---|
| COSINE | Question: Justin has a box that is 12 inches in height. The length of the box is 3 times its height and 4 times its width. What is the volume of the box?
Question: John builds a box. The box is 26 inches by 26 inches by 14 inches. The walls are 1 inch thick on each side. How much is the internal volume in cubic feet? |
| BSR | Question: A window is made up of 8 glass panes. Each pane has a length of 12 inches and a width of 8 inches. What is the area of the window?
Question: John builds a box. The box is 26 inches by 26 inches by 14 inches. The walls are 1 inch thick on each side. How much is the internal volume in cubic feet? |
| SET-BSR | Question: Jazel has 3 sticks. One stick is 3 centimeters long. The second stick is twice as long while the third stick is 1 centimeter shorter than the second stick. What is the total length of Jazel's sticks when they are put together?
Question: John builds a box. The box is 26 inches by 26 inches by 14 inches. The walls are 1 inch thick on each side. How much is the internal volume in cubic feet? |

Table 9: Demonstrations selected by different methods for the GSM8K input: `John has 3 boxes. Each box is 5 inches by 6 inches by 4 inches. The walls are 1 inch thick. What is the total inner volume of all 3 boxes?` We only show the inputs for clarity. Only BSR solves this input (2-shot ICL with Codex). All three methods select one example that demonstrates most of the aspects of the test input, i.e., computing the volume of a box after subtracting wall thickness. The remaining aspect is computing the total of a quantity computed for 3 identical items. COSINE fails to do so, selecting yet another example that requires computing a single box's volume. Since SET-BSR prioritizes coverage of the remaining aspect, it selects an example that has exactly three items whose total length has to be computed but overall is not very similar in reasoning. BSR on the other hand tries to find an example that demonstrates all aspects by itself and happens to find one that partially demonstrates the remaining aspect as well.

| Selector | Prompt |
|---|---|
| BSR | Begun in 1960 and opened to traffic in 1968, the bridge is a two-tiered road and rail design spanning 4,600 metres on the upper deck, with approximately 1,580 metres spanning the river itself. Can we know "What type of design is the bridge?"? Yes

The BBC also introduced Ceefax, the first teletext service, starting in 1974. Can we know "What kind of service was Ceefax?"? Yes

The Water, Sanitation and Hygiene (WSH) program of the Gates Foundation was launched in mid-2005 as a "Learning Initiative," and became a full-fledged program under the Global Development Division in early 2010. Can we know "What was the WSH program launched in 2005"? Yes

Television broadcasting in Hyderabad began in 1974 with the launch of Doordarshan, the Government of India's public service broadcaster, which transmits two free-to-air terrestrial television channelsand one satellite channel. Can we know "What is Doordarshan?"? Yes |

Table 10: Top four demonstrations selected by different methods for the QNLI input: `Telenet was incorporated in 1973 and started operations in 1975. Can we know "What was telenet"?` Since BSR doesn't have access to the labels and also cannot reason about the inputs themselves, it cannot account for the fact that the context in the test input does not contain the answer for the question and selects demonstrations that are all answered "Yes" even though the answer to the test input is "No".

| LM | Dataset Split Selector | ATIS IID/Templ. | Overnight IID/Templ. | Break IID | MTOP IID | GeoQuery IID/Templ./TMCD/Len. | SMCalFlow-CS 8_S/32_C | AVERAGE All/IID/Comp. |
|---|---|---|---|---|---|---|---|---|
| GPT-Neo-2.7B | EPR | 66.1 / 12.2 | 52.3 / 0.9 | 29.9 | 62.2 | 71.4 / 33.6 / 43.6 / 28.8 | 54.5 / 3.6 | 38.3 / 56.1 / 20.5 |
| | CEIL | 67.8 / 18.7 | 50.7 / 2.1 | 29.9 | 60.5 | 65.4 / 30.2 / 43.6 / 25.2 | 59.1 / 3.8 | 38.1 / 55.6 / 20.6 |
| | Random | 12.4 / 0.0 | 3.6 / 0.0 | 1.9 | 1.3 | 17.5 / 11.0 / 14.0 / 0.9 | 3.0 / 0.0 | 5.5 / 6.6 / 4.3 |
| | Cosine | 46.1 / 6.5 | 38.3 / 0.4 | 22.3 | 43.9 | 67.9 / 24.1 / 41.4 / 28.5 | 25.2 / 1.2 | 28.8 / 40.6 / 17.0 |
| | BM25 | 49.5 / 7.4 | 33.7 / 3.0 | 26.5 | 47.7 | 63.6 / 40.6 / 42.1 / 25.5 | 32.0 / 3.2 | 31.2 / 42.2 / 20.3 |
| | BSR | 48.3 / 7.8 | 40.1 / 2.6 | 29.1 | 54.5 | 67.1 / 40.7 / 47.7 / 28.2 | 39.7 / 3.5 | 34.1 / 46.5 / 21.7 |
| | Set-BSR | 54.6 / 13.2 | 43.2 / 4.9 | 28.6 | 55.1 | 67.1 / 45.3 / 45.4 / 26.4 | 41.5 / 4.8 | 35.8 / 48.4 / 23.3 |
| LLaMA-7B | EPR | 73.0 / 21.0 | 57.7 / 1.8 | 33.2 | 65.2 | 75.4 / 49.3 / 45.8 / 30.3 | 64.0 / 8.0 | 43.7 / 61.4 / 26.0 |
| | CEIL | 74.0 / 30.5 | 55.8 / 4.4 | 36.1 | 66.8 | 66.8 / 50.5 / 45.3 / 24.2 | 67.4 / 11.9 | 44.5 / 61.1 / 27.8 |
| | Random | 9.5 / 0.0 | 4.2 / 0.5 | 8.8 | 2.8 | 9.3 / 13.3 / 9.2 / 4.5 | 6.2 / 0.0 | 5.7 / 6.8 / 4.6 |
| | Cosine | 56.7 / 11.5 | 48.7 / 0.0 | 26.1 | 49.8 | 73.9 / 33.5 / 42.6 / 29.4 | 37.3 / 3.2 | 34.4 / 48.8 / 20.0 |
| | BM25 | 61.0 / 12.5 | 45.1 / 2.5 | 30.1 | 53.6 | 67.9 / 39.5 / 44.9 / 30.6 | 43.4 / 9.5 | 36.7 / 50.2 / 23.3 |
| | BSR | 60.9 / 14.3 | 51.2 / 3.0 | 32.5 | 59.1 | 72.5 / 47.2 / 46.9 / 30.3 | 54.1 / 9.8 | 40.1 / 55.0 / 25.2 |
| | Set-BSR | 64.3 / 21.2 | 51.5 / 6.3 | 33.7 | 61.9 | 76.1 / 52.3 / 48.5 / 35.8 | 54.5 / 19.3 | 43.8 / 57.0 / 30.6 |
| LLaMA-13B | EPR | 75.3 / 28.2 | 62.6 / 0.7 | 37.2 | 68.3 | 80.4 / 57.5 / 53.9 / 34.8 | 66.5 / 11.9 | 48.1 / 65.0 / 31.2 |
| | CEIL | 76.1 / 36.7 | 59.1 / 4.9 | 38.8 | 71.2 | 75.7 / 59.3 / 55.5 / 32.1 | 68.3 / 21.0 | 49.9 / 64.9 / 34.9 |
| | Random | 19.5 / 5.1 | 3.4 / 2.6 | 9.0 | 4.8 | 23.9 / 19.8 / 14.2 / 1.5 | 14.0 / 0.0 | 9.8 / 12.4 / 7.2 |
| | Cosine | 57.8 / 20.5 | 48.6 / 3.0 | 29.4 | 54.0 | 77.1 / 44.8 / 48.5 / 32.7 | 42.9 / 5.4 | 38.7 / 51.6 / 25.8 |
| | BM25 | 65.6 / 22.6 | 50.3 / 5.8 | 34.6 | 58.7 | 76.4 / 49.3 / 50.3 / 37.6 | 48.2 / 13.6 | 42.8 / 55.6 / 29.9 |
| | BSR | 64.0 / 22.8 | 55.7 / 5.8 | 37.7 | 64.4 | 79.6 / 60.1 / 55.2 / 38.5 | 60.1 / 14.5 | 46.5 / 60.3 / 32.8 |
| | Set-BSR | 69.6 / 33.5 | 59.6 / 8.6 | 39.2 | 66.1 | 81.8 / 64.1 / 60.2 / 44.8 | 62.5 / 26.7 | 51.4 / 63.1 / 39.7 |
| StarCoder | EPR | 75.8 / 42.9 | 66.2 / 4.6 | 41.5 | 72.6 | 85.4 / 64.4 / 59.6 / 42.1 | 69.8 / 17.3 | 53.5 / 68.5 / 38.5 |
| | CEIL | 79.2 / 49.7 | 64.6 / 17.6 | 42.5 | 73.7 | 85.0 / 70.9 / 61.8 / 39.7 | 71.0 / 31.8 | 57.3 / 69.3 / 45.3 |
| | Random | 22.2 / 12.0 | 10.8 / 3.3 | 8.8 | 4.7 | 25.7 / 27.9 / 21.5 / 6.1 | 20.1 / 0.0 | 13.6 / 15.4 / 11.8 |
| | Cosine | 68.9 / 31.9 | 63.8 / 11.1 | 31.5 | 62.1 | 81.4 / 57.9 / 54.4 / 42.7 | 48.8 / 10.7 | 47.1 / 59.4 / 34.8 |
| | BM25 | 70.9 / 40.4 | 62.3 / 17.6 | 37.0 | 66.8 | 85.0 / 64.9 / 60.1 / 47.3 | 57.7 / 24.3 | 52.9 / 63.3 / 42.4 |
| | BSR | 72.1 / 38.3 | 63.5 / 16.4 | 39.4 | 69.2 | 85.4 / 72.0 / 62.6 / 45.5 | 66.3 / 27.5 | 54.8 / 66.0 / 43.7 |
| | Set-BSR | 78.2 / 50.6 | 67.3 / 27.1 | 39.7 | 73.4 | 86.8 / 76.1 / 64.8 / 54.5 | 70.7 / 50.1 | 61.6 / 69.3 / 53.9 |
| Cushman | EPR | 75.1 / 41.1 | 63.3 / 3.3 | 37.5 | 70.8 | 85.0 / 63.4 / 57.4 / 40.3 | 68.7 / 16.0 | 51.8 / 66.7 / 36.9 |
| | CEIL | 76.8 / 49.2 | 63.0 / 11.4 | 36.2 | 70.9 | 82.5 / 66.4 / 60.6 / 40.3 | 69.9 / 30.0 | 54.8 / 66.6 / 43.0 |
| | Random | 16.9 / 11.1 | 5.1 / 2.8 | 13.0 | 6.8 | 22.1 / 24.9 / 15.7 / 5.5 | 20.1 / 0.0 | 12.0 / 14.0 / 10.0 |
| | Cosine | 64.6 / 32.5 | 60.7 / 6.2 | 30.6 | 59.7 | 81.1 / 56.1 / 50.3 / 42.1 | 46.7 / 8.0 | 44.9 / 57.2 / 32.5 |
| | BM25 | 67.1 / 35.1 | 58.3 / 11.8 | 34.5 | 64.4 | 81.4 / 61.6 / 56.8 / 47.9 | 56.2 / 21.0 | 49.7 / 60.3 / 39.0 |
| | BSR | 69.4 / 36.3 | 61.0 / 10.9 | 38.3 | 68.4 | 83.9 / 67.4 / 61.4 / 46.7 | 62.4 / 24.7 | 52.6 / 63.9 / 41.2 |
| | Set-BSR | 76.7 / 46.9 | 63.9 / 19.9 | 40.2 | 71.5 | 88.6 / 74.9 / 64.1 / 50.3 | 68.7 / 47.8 | 59.5 / 68.3 / 50.7 |
| GPT-3.5-Turbo | EPR | 74.5 / 41.8 | 56.8 / 10.6 | 0.0 | 64.8 | 82.1 / 54.6 / 56.3 / 39.4 | 66.9 / 21.0 | 47.4 / 57.5 / 37.3 |
| | CEIL | 79.2 / 53.4 | 54.9 / 26.9 | 0.0 | 68.2 | 75.4 / 56.9 / 55.8 / 33.0 | 72.8 / 37.6 | 51.2 / 58.4 / 43.9 |
| | Random | 19.8 / 14.1 | 8.4 / 2.5 | 14.5 | 4.8 | 26.8 / 27.2 / 17.8 / 3.3 | 17.1 / 0.0 | 13.0 / 15.2 / 10.8 |
| | Cosine | 64.2 / 32.8 | 58.1 / 12.0 | 30.2 | 54.6 | 76.1 / 56.5 / 55.1 / 39.1 | 48.8 / 11.2 | 44.9 / 55.3 / 34.4 |
| | BM25 | 71.6 / 39.5 | 52.5 / 16.9 | 36.3 | 60.5 | 78.9 / 63.3 / 58.1 / 41.2 | 59.8 / 24.6 | 50.3 / 59.9 / 40.6 |
| | BSR | 73.2 / 39.3 | 55.6 / 17.6 | 38.6 | 63.2 | 82.1 / 69.7 / 56.3 / 40.3 | 66.5 / 29.7 | 52.7 / 63.2 / 42.1 |
| | Set-BSR | 78.6 / 53.3 | 58.2 / 25.9 | 39.5 | 66.9 | 83.9 / 74.1 / 61.3 / 52.4 | 71.6 / 55.1 | 60.1 / 66.5 / 53.7 |
| Codex | EPR | 80.0 / 49.0 | 69.6 / 12.1 | 45.5 | 76.6 | 87.9 / 70.9 / 64.5 / 47.6 | 76.3 / 21.7 | 58.5 / 72.6 / 44.3 |
| | CEIL | 83.6 / 60.3 | 69.8 / 29.6 | 46.5 | 79.7 | 85.4 / 76.6 / 68.8 / 50.3 | 77.5 / 40.1 | 64.0 / 73.7 / 54.3 |
| | Random | 27.3 / 14.3 | 16.1 / 6.5 | 26.7 | 9.3 | 36.4 / 28.3 / 25.7 / 19.1 | 31.9 / 7.4 | 20.7 / 24.6 / 16.9 |
| | Cosine | 74.1 / 39.5 | 67.7 / 17.1 | 41.0 | 69.1 | 86.8 / 68.4 / 61.5 / 48.5 | 54.7 / 11.9 | 53.4 / 65.6 / 41.1 |
| | BM25 | 76.9 / 46.7 | 66.5 / 30.1 | 45.2 | 72.3 | 87.1 / 77.2 / 71.2 / 62.1 | 65.4 / 29.4 | 60.9 / 68.9 / 52.8 |
| | BSR | 79.3 / 48.1 | 68.1 / 22.4 | 44.0 | 76.8 | 88.2 / 78.8 / 71.6 / 53.0 | 72.5 / 31.5 | 61.2 / 71.5 / 50.9 |
| | Set-BSR | 84.7 / 62.4 | 69.5 / 41.4 | 46.0 | 79.6 | 91.1 / 86.6 / 76.0 / 69.4 | 75.7 / 61.2 | 70.3 / 74.4 / 66.2 |

Table 11: Comparison of various methods on 8-shot ICL for semantic parsing datasets. Right-most column shows average performances on ALL, only IID, and only COMPositional splits. BSR consistently outperforms COSINE and BM25 and SET-BSR yields further gains surpassing even trained methods.

| LM | Selector | GSM8K | DROP | QNLI | MNLI | RTE | MRPC | PAWS | QQP | SST2 | AVERAGE |
|---|---|---|---|---|---|---|---|---|---|---|---|
| **GPT-Neo-2.7B** | **Random** | 1.5 | 6.4 | 51.2 | 41.9 | 53.4 | 51 | 48 | 65.9 | 86.9 | 45.1 |
| | **Cosine** | 1.9 | 12.3 | 56 | 44 | 54.2 | 52.5 | 52.5 | 75 | 81.9 | 47.8 |
| | **BM25** | 3.7 | 12.6 | 58.1 | 42.2 | 50.9 | 57.6 | 55.2 | 71.3 | 82.6 | 48.2 |
| | **BSR** | 2.9 | 13.9 | 81.1 | 76.7 | 67.9 | 70.1 | 75 | 86.4 | 90.9 | 62.8 |
| | **Set-BSR** | 2.9 | 13.2 | 78.6 | 61.5 | 60.6 | 68.4 | 74.9 | 84.4 | 89.8 | 59.4 |
| **LLaMA-7B** | **Random** | 11.3 | 23.4 | 54.2 | 54.3 | 70 | 33.8 | 59.1 | 66.2 | 94.2 | 51.8 |
| | **Cosine** | 12.3 | 26.7 | 58 | 58 | 67.9 | 46.6 | 56.6 | 76.1 | 92 | 54.9 |
| | **BM25** | 11.9 | 26.8 | 57.3 | 56.1 | 68.2 | 48.3 | 57.2 | 73.2 | 93.2 | 54.7 |
| | **BSR** | 14.5 | 27.2 | 82.9 | 76.3 | 70.8 | 59.8 | 74 | 80.4 | 95.8 | 64.6 |
| | **Set-BSR** | 15 | 29.1 | 74.7 | 70.2 | 67.5 | 53.7 | 72 | 80.3 | 94.2 | 61.9 |
| **LLaMA-13B** | **Random** | 15.3 | 29.9 | 56.3 | 51.3 | 75.5 | 72.3 | 60.1 | 67.3 | 93.1 | 57.9 |
| | **Cosine** | 16.7 | 32.6 | 61.8 | 62.9 | 75.1 | 57.1 | 58.9 | 78.7 | 92.7 | 59.6 |
| | **BM25** | 16.3 | 31.7 | 63.3 | 62.8 | 73.6 | 62.5 | 58.8 | 76.9 | 92.7 | 59.8 |
| | **BSR** | 19.2 | 34.3 | 85.6 | 82 | 76.5 | 72.8 | 77.5 | 85.2 | 95.2 | 69.8 |
| | **Set-BSR** | 20.6 | 33.3 | 82 | 73.3 | 76.2 | 67.2 | 77.2 | 81.9 | 93.8 | 67.3 |

Table 12: Comparison of various methods on 8-shot ICL for reasoning and classification datasets. BSR outperforms all prior training-free methods though SET-BSR doesn't yield additional improvement.

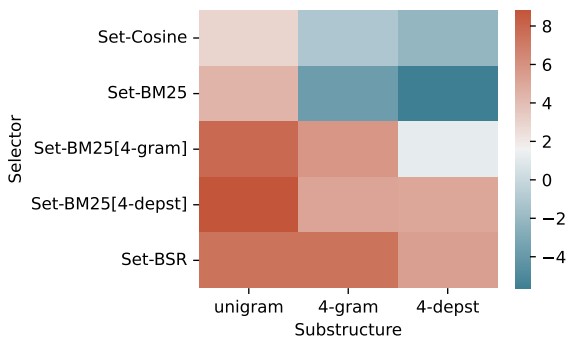

Figure 11: **Coverage of aspects of the test instance:** Change in recall of unigrams, 4-grams, and dependency parse subtrees (size < 4) in the test input with set-selection of demonstrations, compared to their non-set version, averaged across all datasets.

| LM | Selector | GSM8K | DROP | QNLI |
|---|---|---|---|---|
| **GPT-Neo-2.7B** | **EPR** | 0.0 | 13.8 | 74.9 |
| | **CEIL** | 0.0 | - | 84.2 |
| | **BSR** | 2.9 | 13.9 | 81.1 |
| | **Set-BSR** | 2.9 | 13.2 | 78.6 |
| **LLaMA-7B** | **EPR** | - | - | 80.9 |
| | **CEIL** | - | - | 84.1 |
| | **BSR** | 14.5 | 27.2 | 82.9 |
| | **Set-BSR** | 15.0 | 29.1 | 74.7 |
| **LLaMA-13B** | **EPR** | - | - | 81.9 |
| | **CEIL** | - | - | 84.2 |
| | **BSR** | 19.2 | 34.3 | 85.6 |
| | **Set-BSR** | 20.6 | 33.3 | 82.0 |
| **StarCoder** | **EPR** | - | - | 81.7 |
| | **CEIL** | - | - | 84.8 |
| | **BSR** | 17.1 | 26.6 | 84.7 |
| | **Set-BSR** | 17.5 | 24.9 | 80.3 |
| **Cushman** | **EPR** | 10.0 | - | 78.9 |
| | **CEIL** | 8.3 | - | 83.8 |
| | **BSR** | 12.1 | 23.6 | 84.7 |
| | **Set-BSR** | 11.1 | 23.7 | 74.4 |
| **Codex** | **EPR** | 61.7 | - | 83.8 |
| | **CEIL** | 63.1 | - | 84.9 |
| | **BSR** | 68.1 | 68.1 | 88.7 |
| | **Set-BSR** | 67.4 | 66.4 | 84.6 |

Table 13: Additional non-semantic parsing 8-shot ICL results for comparison with trained methods and larger LLMs. BSR is competitive with EPR and CEIL, even outperforming them with larger LLMs. We could not get CEIL to work for DROP.

| | GPT-Neo-2.7B | LLaMA-7B | LLaMA-13B |
|---|---|---|---|
| **Random** | 22.5 | 25.5 | 30.4 |
| **Cosine** | 36.9 | 43.2 | 47.7 |
| **BM25** | 38.5 | 44.4 | 50.1 |
| **BSR** | 46.4 | 50.6 | 56.5 |
| **Set-BSR** | 45.9 | 51.6 | 58.2 |

Table 14: Average 8-shot ICL performance across all datasets and splits.

| Dataset
Split
Selector | ATIS
IID / Tpl. | Overnight
IID / Tpl. | Break
IID | MTOP
IID | GeoQuery
IID / Tpl. / TMCD / Len. | SMCalFlow-CS
8_S / 32_C | AVERAGE
All / IID / Comp. |
|---|---|---|---|---|---|---|---|
| **Random** | 12.4 / 0.0 | 3.6 / 0.0 | 1.9 | 1.3 | 17.5 / 11.0 / 14.0 / 0.9 | 3.0 / 0.0 | 5.5 / 6.6 / 4.3 |
| **Cosine[bert-base]** | 43.6 / 3.4 | 31.8 / 0.9 | 22.4 | 44.1 | 67.9 / 37.9 / 41.4 / 25.5 | 24.3 / 0.9 | 28.7 / 39.0 / 18.3 |
| **Cosine[mpnet-base]** | 46.1 / 6.5 | 38.3 / 0.4 | 22.3 | 43.9 | 67.9 / 24.1 / 41.4 / 28.5 | 25.2 / 1.2 | 28.8 / 40.6 / 17.0 |
| **+ Coverage** | 45.9 / 4.2 | 32.6 / 0.4 | 19.2 | 39.2 | 59.6 / 32.6 / 44.0 / 18.8 | 27.2 / 1.4 | 27.1 / 37.3 / 16.9 |
| **- reorder** | 46.5 / 6.5 | 32.3 / 0.5 | 19.4 | 37.8 | 58.9 / 32.3 / 43.1 / 20.6 | 27.2 / 1.1 | 27.2 / 37.0 / 17.4 |
| **BM25** | 49.5 / 7.4 | 33.7 / 3.0 | 26.5 | 47.7 | 63.6 / 40.6 / 42.1 / 25.5 | 32.0 / 3.2 | 31.2 / 42.2 / 20.3 |
| **+ Coverage** | 44.8 / 8.1 | 25.8 / 0.4 | 21.1 | 41.5 | 56.8 / 27.0 / 39.8 / 20.9 | 22.1 / 1.5 | 25.8 / 35.3 / 16.3 |
| **- reorder** | 44.9 / 7.4 | 24.1 / 0.4 | 20.7 | 40.6 | 53.6 / 28.3 / 39.3 / 20.0 | 23.0 / 2.3 | 25.4 / 34.5 / 16.3 |
| **BM25[4-gram]** | 45.4 / 5.5 | 33.4 / 2.1 | 27.3 | 48.9 | 65.4 / 39.6 / 39.6 / 27.3 | 28.4 / 3.5 | 30.5 / 41.5 / 19.6 |
| **+ Coverage** | 48.3 / 8.5 | 27.2 / 2.3 | 23.8 | 47.1 | 63.2 / 36.2 / 42.6 / 24.2 | 26.6 / 3.9 | 29.5 / 39.4 / 19.6 |
| **- reorder** | 47.1 / 8.1 | 27.1 / 1.8 | 20.9 | 46.1 | 59.3 / 30.8 / 39.2 / 24.5 | 25.7 / 2.7 | 27.8 / 37.7 / 17.9 |
| **BM25[4-depst]** | 45.1 / 9.0 | 31.8 / 1.9 | 26.2 | 47.4 | 67.1 / 38.7 / 43.6 / 24.2 | 27.5 / 1.7 | 30.4 / 40.9 / 19.9 |
| **+ Coverage** | 48.5 / 12.0 | 28.7 / 0.9 | 22.8 | 44.7 | 58.6 / 38.8 / 41.8 / 24.2 | 26.6 / 2.4 | 29.2 / 38.3 / 20.0 |
| **- reorder** | 47.1 / 9.3 | 28.7 / 1.1 | 21.9 | 46.1 | 59.6 / 34.1 / 34.7 / 29.1 | 25.5 / 2.0 | 28.3 / 38.2 / 18.4 |
| **BSR[bert-base]** | 50.1 / 8.5 | 36.9 / 1.9 | 27.9 | 51.5 | 67.9 / 46.2 / 45.5 / 27.0 | 35.8 / 2.4 | 33.5 / 45.0 / 21.9 |
| **+ Coverage** | 53.6 / 12.2 | 38.4 / 2.3 | 28.3 | 52.1 | 70.0 / 46.9 / 48.5 / 27.6 | 39.7 / 4.5 | 35.3 / 47.0 / 23.7 |
| **BSF1[deberta-base]** | 47.2 / 6.0 | 39.4 / 1.4 | 30.6 | 54.8 | 66.4 / 41.0 / 44.5 / 29.7 | 41.4 / 1.8 | 33.7 / 46.6 / 20.7 |
| **BSP[deberta-base]** | 45.2 / 7.2 | 39.4 / 1.9 | 31.1 | 46.5 | 64.6 / 38.1 / 41.4 / 27.6 | 27.5 / 0.9 | 31.0 / 42.4 / 19.5 |
| **BSR[deberta-base]** | 47.5 / 9.0 | 37.2 / 2.5 | 30.2 | 53.5 | 68.9 / 44.5 / 44.4 / 27.3 | 40.6 / 4.1 | 34.1 / 46.3 / 22.0 |
| **+ IDF** | 48.2 / 9.3 | 40.0 / 2.1 | 28.2 | 53.7 | 68.9 / 40.2 / 42.7 / 31.5 | 39.1 / 4.1 | 34.0 / 46.4 / 21.7 |
| **+ Coverage** | 53.8 / 14.8 | 40.1 / 2.6 | 28.5 | 55.4 | 74.6 / 45.3 / 45.4 / 24.2 | 40.3 / 4.1 | 35.8 / 48.8 / 22.7 |
| **+ IDF** | 54.1 / 12.3 | 40.6 / 3.3 | 28.3 | 53.9 | 72.1 / 46.5 / 41.6 / 29.7 | nan / 3.6 | 35.1 / 49.8 / 22.9 |
| **- reorder** | 50.0 / 11.8 | 37.7 / 2.1 | 26.9 | 53.0 | 66.8 / 44.5 / 42.9 / 24.8 | 39.0 / 4.7 | 33.7 / 45.6 / 21.8 |
| **BSR[deberta-large]** | 48.3 / 7.8 | 40.1 / 2.6 | 29.1 | 54.5 | 67.1 / 40.7 / 47.7 / 28.2 | 39.7 / 3.5 | 34.1 / 46.5 / 21.7 |
| **+ IDF** | 48.8 / 9.7 | 41.5 / 2.6 | 27.8 | 54.0 | 69.3 / 36.6 / 45.4 / 26.4 | 40.3 / 3.8 | 33.8 / 47.0 / 20.7 |
| **+ Coverage** | 54.6 / 13.2 | 43.2 / 4.9 | 28.6 | 55.1 | 67.1 / 45.3 / 45.4 / 26.4 | 41.5 / 4.8 | 35.8 / 48.4 / 23.3 |
| **+ IDF** | nan / nan | nan / nan | 29.0 | nan | 72.1 / 43.8 / 46.8 / 25.8 | 40.9 / 5.1 | 37.6 / 47.4 / 30.4 |
| **- reorder** | 52.0 / 12.0 | 37.6 / 4.8 | 29.2 | 53.5 | 63.9 / 39.9 / 44.4 / 24.2 | 39.0 / 5.1 | 33.8 / 45.9 / 21.7 |

Table 15: 8-shot ICL results with GPT-Neo-2.7B for all ablations of learning-free methods on semantic parsing datasets and splits.

| Dataset
Split
Selector | ATIS
IID / Tpl. | Overnight
IID / Tpl. | Break
IID | MTOP
IID | GeoQuery
IID / Tpl. / TMCD / Len. | SMCalFlow-CS
8_S / 32_C | AVERAGE
All / IID / Comp. |
|---|---|---|---|---|---|---|---|
| **Random** | 9.5 / 0.0 | 4.2 / 0.5 | 8.8 | 2.8 | 9.3 / 13.3 / 9.2 / 4.5 | 6.2 / 0.0 | 5.7 / 6.8 / 4.6 |
| **Cosine[bert-base]** | 52.9 / 7.2 | 38.5 / 2.8 | 24.9 | 49.0 | 69.6 / 40.7 / 42.4 / 29.4 | 35.2 / 1.7 | 32.9 / 45.0 / 20.7 |
| **Cosine[mpnet-base]** | 56.7 / 11.5 | 48.7 / 0.0 | 26.1 | 49.8 | 73.9 / 33.5 / 42.6 / 29.4 | 37.3 / 3.2 | 34.4 / 48.8 / 20.0 |
| **+ Coverage** | 55.0 / 13.9 | 42.5 / 0.9 | 24.7 | 43.0 | 63.9 / 37.6 / 43.7 / 29.7 | 35.2 / 3.3 | 32.8 / 44.1 / 21.5 |
| **- reorder** | 56.0 / 14.3 | 41.8 / 1.2 | 25.5 | 43.6 | 65.4 / 35.8 / 43.6 / 35.2 | 35.2 / 3.5 | 33.4 / 44.6 / 22.3 |
| **BM25** | 61.0 / 12.5 | 45.1 / 2.5 | 30.1 | 53.6 | 67.9 / 39.5 / 44.9 / 30.6 | 43.4 / 9.5 | 36.7 / 50.2 / 23.3 |
| **+ Coverage** | 55.7 / 12.3 | 34.9 / 2.6 | 25.5 | 48.5 | 63.6 / 30.5 / 40.8 / 25.5 | 35.3 / 6.2 | 31.8 / 43.9 / 19.7 |
| **- reorder** | 55.0 / 12.5 | 34.4 / 2.8 | 25.1 | 48.3 | 60.4 / 31.9 / 41.8 / 22.7 | 36.3 / 6.0 | 31.4 / 43.2 / 19.6 |
| **BM25[4-gram]** | 55.5 / 10.8 | 42.4 / 5.6 | 31.9 | 52.2 | 70.7 / 44.7 / 41.7 / 30.9 | 41.4 / 9.5 | 36.4 / 49.0 / 23.9 |
| **+ Coverage** | 56.7 / 16.2 | 39.6 / 5.8 | 29.4 | 53.5 | 68.6 / 42.9 / 42.6 / 34.5 | 45.8 / 13.6 | 37.4 / 48.9 / 25.9 |
| **- reorder** | 57.3 / 16.0 | 37.0 / 4.9 | 28.0 | 51.4 | 68.9 / 41.0 / 44.1 / 35.8 | 44.9 / 13.4 | 36.9 / 47.9 / 25.9 |
| **BM25[4-depst]** | 54.2 / 11.1 | 42.3 / 3.3 | 29.7 | 52.2 | 68.9 / 41.9 / 43.4 / 28.5 | 39.9 / 5.4 | 35.1 / 47.9 / 22.3 |
| **+ Coverage** | 59.7 / 17.8 | 41.1 / 5.6 | 29.2 | 52.4 | 63.6 / 38.9 / 40.9 / 37.3 | 45.6 / 11.5 | 37.0 / 48.6 / 25.3 |
| **- reorder** | 59.6 / 15.3 | 39.9 / 4.6 | 28.1 | 51.2 | 66.4 / 39.9 / 44.2 / 37.0 | 44.1 / 10.7 | 36.8 / 48.2 / 25.3 |
| **BSR[bert-base]** | 63.0 / 15.0 | 48.5 / 2.6 | 32.7 | 57.4 | 73.9 / 50.5 / 45.8 / 30.0 | 50.9 / 8.4 | 39.9 / 54.4 / 25.4 |
| **+ Coverage** | 64.3 / 17.8 | 48.3 / 5.8 | 34.1 | 59.2 | 76.8 / 51.9 / 48.7 / 34.8 | 50.2 / 17.6 | 42.5 / 55.5 / 29.4 |
| **BSF1[deberta-base]** | 58.8 / 13.1 | 49.7 / 3.5 | 33.4 | 59.5 | 75.7 / 41.9 / 43.8 / 32.7 | 55.7 / 8.4 | 39.7 / 55.5 / 23.9 |
| **BSP[deberta-base]** | 53.5 / 9.9 | 49.2 / 3.7 | 32.2 | 47.9 | 77.1 / 42.4 / 43.8 / 35.2 | 41.7 / 2.9 | 36.6 / 50.3 / 23.0 |
| **BSR[deberta-base]** | 61.2 / 15.2 | 50.3 / 2.6 | 33.4 | 59.1 | 74.3 / 50.2 / 45.9 / 33.6 | 51.2 / 11.6 | 40.7 / 54.9 / 26.5 |
| **+ IDF** | 60.8 / 15.3 | 51.4 / 4.2 | 32.4 | 58.4 | 70.7 / 47.5 / 42.0 / 35.2 | 50.3 / 12.1 | 40.0 / 54.0 / 26.1 |
| **+ Coverage** | 62.5 / 18.3 | 52.5 / 5.5 | 34.3 | 60.0 | 75.4 / 45.5 / 48.4 / 37.0 | 53.0 / 17.5 | 42.5 / 56.3 / 28.7 |
| **+ IDF** | 63.2 / 19.4 | 51.1 / 6.2 | 32.8 | 60.2 | 74.6 / 45.3 / 43.1 / 33.9 | 53.3 / 17.5 | 41.7 / 55.9 / 27.6 |
| **- reorder** | 63.7 / 20.6 | 49.6 / 6.2 | 35.3 | 61.7 | 73.9 / 47.3 / 48.4 / 34.2 | 52.4 / 17.5 | 42.6 / 56.1 / 29.0 |
| **BSR[deberta-large]** | 60.9 / 14.3 | 51.2 / 3.0 | 32.5 | 59.1 | 72.5 / 47.2 / 46.9 / 30.3 | 54.1 / 9.8 | 40.1 / 55.0 / 25.2 |
| **+ IDF** | 61.6 / 15.0 | 51.4 / 4.6 | 33.2 | 59.1 | 73.2 / 53.2 / 46.5 / 32.1 | 52.0 / 10.7 | 41.0 / 55.1 / 27.0 |
| **+ Coverage** | 64.3 / 21.2 | 51.5 / 6.3 | 33.7 | 61.9 | 76.1 / 52.3 / 48.5 / 35.8 | 54.5 / 19.3 | 43.8 / 57.0 / 30.6 |
| **+ IDF** | 65.6 / 21.9 | 51.5 / 6.2 | 32.9 | 61.5 | 77.1 / 52.8 / 50.0 / 35.8 | 55.6 / 18.6 | 44.1 / 57.4 / 30.9 |
| **- reorder** | 65.4 / 19.6 | 51.6 / 7.0 | 33.5 | 61.9 | 76.1 / 51.2 / 48.6 / 35.5 | 54.5 / 19.6 | 43.7 / 57.2 / 30.2 |

Table 16: 8-shot ICL results with LLaMA-7B for all ablations of learning-free methods on semantic parsing datasets and splits.

| Dataset
Split
Selector | ATIS
IID / Tpl. | Overnight
IID / Tpl. | Break
IID | MTOP
IID | GeoQuery
IID / Tpl. / TMCD / Len. | SMCalFlow-CS
8_S / 32_C | AVERAGE
All / IID / Comp. |
|---|---|---|---|---|---|---|---|
| **Random** | 19.5 / 5.1 | 3.4 / 2.6 | 9.0 | 4.8 | 23.9 / 19.8 / 14.2 / 1.5 | 14.0 / 0.0 | 9.8 / 12.4 / 7.2 |
| **Cosine[bert-base]** | 55.8 / 17.8 | 44.7 / 5.1 | 30.7 | 52.7 | 75.7 / 53.1 / 49.2 / 31.5 | 42.7 / 3.0 | 38.5 / 50.4 / 26.6 |
| **Cosine[mpnet-base]** | 57.8 / 20.5 | 48.6 / 3.0 | 29.4 | 54.0 | 77.1 / 44.8 / 48.5 / 32.7 | 42.9 / 5.4 | 38.7 / 51.6 / 25.8 |
| **+ Coverage** | 57.5 / 23.8 | 42.2 / 5.8 | 28.0 | 50.2 | 68.6 / 51.9 / 51.0 / 30.0 | 41.8 / 5.7 | 38.0 / 48.1 / 28.0 |
| **- reorder** | 56.8 / 21.9 | 41.6 / 6.0 | 27.6 | 49.9 | 68.9 / 54.2 / 51.0 / 31.8 | 39.6 / 5.4 | 37.9 / 47.4 / 28.4 |
| **BM25** | 65.6 / 22.6 | 50.3 / 5.8 | 34.6 | 58.7 | 76.4 / 49.3 / 50.3 / 37.6 | 48.2 / 13.6 | 42.8 / 55.6 / 29.9 |
| **+ Coverage** | 59.3 / 21.2 | 37.5 / 3.7 | 29.3 | 52.7 | 67.1 / 46.6 / 49.1 / 35.2 | 40.6 / 8.6 | 37.6 / 47.8 / 27.4 |
| **- reorder** | 57.2 / 20.6 | 37.5 / 3.3 | 28.5 | 53.5 | 67.1 / 43.6 / 47.3 / 29.7 | 40.3 / 8.7 | 36.5 / 47.4 / 25.6 |
| **BM25[4-gram]** | 58.6 / 20.3 | 49.7 / 7.0 | 33.9 | 58.1 | 77.9 / 55.5 / 52.0 / 34.8 | 50.6 / 13.0 | 42.6 / 54.8 / 30.4 |
| **+ Coverage** | 62.4 / 26.5 | 45.6 / 6.7 | 31.8 | 57.5 | 73.6 / 55.9 / 51.5 / 40.6 | 49.1 / 20.4 | 43.5 / 53.3 / 33.6 |
| **- reorder** | 60.8 / 26.5 | 44.8 / 6.5 | 32.4 | 56.7 | 76.1 / 56.4 / 51.0 / 40.6 | 50.6 / 19.9 | 43.5 / 53.6 / 33.5 |
| **BM25[4-depst]** | 58.4 / 23.8 | 48.6 / 6.2 | 32.9 | 59.2 | 75.4 / 55.4 / 52.6 / 41.5 | 46.4 / 6.3 | 42.2 / 53.5 / 31.0 |
| **+ Coverage** | 63.7 / 28.4 | 45.8 / 7.2 | 32.1 | 58.8 | 75.7 / 55.3 / 50.6 / 41.5 | 50.5 / 19.8 | 44.1 / 54.4 / 33.8 |
| **- reorder** | 64.5 / 28.9 | 43.9 / 6.7 | 31.2 | 58.5 | 77.1 / 56.8 / 52.8 / 38.5 | 51.5 / 20.1 | 44.2 / 54.5 / 34.0 |
| **BSR[bert-base]** | 65.0 / 27.5 | 53.3 / 5.8 | 35.0 | 63.8 | 77.9 / 59.5 / 54.0 / 39.4 | 55.6 / 13.6 | 45.9 / 58.4 / 33.3 |
| **+ Coverage** | 69.4 / 33.3 | 56.9 / 8.8 | 37.1 | 64.4 | 80.4 / 64.7 / 57.7 / 43.0 | 57.3 / 23.7 | 49.7 / 60.9 / 38.5 |
| **BSF1[deberta-base]** | 62.2 / 23.3 | 55.5 / 4.4 | 35.8 | 59.4 | 80.7 / 59.2 / 52.0 / 37.6 | 59.8 / 17.5 | 45.6 / 58.9 / 32.3 |
| **BSP[deberta-base]** | 56.5 / 16.9 | 55.0 / 4.9 | 34.1 | 49.4 | 80.7 / 54.7 / 50.0 / 33.6 | 46.1 / 5.1 | 40.6 / 53.6 / 27.6 |
| **BSR[deberta-base]** | 63.3 / 26.1 | 56.4 / 6.7 | 36.3 | 64.3 | 80.0 / 61.9 / 54.0 / 40.0 | 59.5 / 15.8 | 47.0 / 60.0 / 34.1 |
| **+ IDF** | 63.8 / 28.0 | 53.8 / 5.6 | 36.6 | 64.3 | 77.5 / 58.2 / 54.7 / 34.8 | 55.3 / 17.9 | 45.9 / 58.5 / 33.2 |
| **+ Coverage** | 69.5 / 32.6 | 58.4 / 9.7 | 36.6 | 66.6 | 80.7 / 61.5 / 56.4 / 43.0 | 63.0 / 26.7 | 50.4 / 62.5 / 38.3 |
| **+ IDF** | 68.6 / 35.4 | 56.8 / 9.5 | 36.8 | 66.6 | 78.6 / 59.4 / 54.8 / 41.5 | nan / 26.8 | 48.6 / 61.5 / 37.9 |
| **- reorder** | 68.9 / 33.0 | 56.5 / 9.5 | 36.7 | 67.5 | 79.6 / 63.7 / 56.5 / 37.3 | 61.9 / 30.9 | 50.2 / 61.9 / 38.5 |
| **BSR[deberta-large]** | 64.0 / 22.8 | 55.7 / 5.8 | 37.7 | 64.4 | 79.6 / 60.1 / 55.2 / 38.5 | 60.1 / 14.5 | 46.5 / 60.3 / 32.8 |
| **+ IDF** | 65.5 / 25.6 | 54.1 / 5.3 | 35.9 | 65.0 | 79.6 / 59.6 / 55.1 / 34.2 | 58.8 / 17.6 | 46.4 / 59.8 / 32.9 |
| **+ Coverage** | 69.6 / 33.5 | 59.6 / 8.6 | 39.2 | 66.1 | 81.8 / 64.1 / 60.2 / 44.8 | 62.5 / 26.7 | 51.4 / 63.1 / 39.7 |
| **+ IDF** | 69.2 / 34.9 | 57.0 / 7.7 | 38.2 | 66.3 | 80.4 / 58.1 / 58.0 / 43.3 | nan / 28.1 | 49.2 / 62.2 / 38.4 |
| **- reorder** | 71.1 / 33.5 | 57.2 / 9.2 | 37.3 | 67.6 | 81.8 / 60.1 / 57.5 / 39.7 | 60.6 / 29.1 | 50.4 / 62.6 / 38.2 |

Table 17: 8-shot ICL results with LLaMA-13B for all ablations of learning-free methods on semantic parsing datasets and splits.

| Dataset
Split
Selector | ATIS
IID / Tpl. | Overnight
IID / Tpl. | Break
IID | MTOP
IID | GeoQuery
IID / Tpl. / TMCD / Len. | SMCalFlow-CS
8_S / 32_C | AVERAGE
All / IID / Comp. |
|---|---|---|---|---|---|---|---|
| **Random** | 22.2 / 12.0 | 10.8 / 3.3 | 8.8 | 4.7 | 25.7 / 27.9 / 21.5 / 6.1 | 20.1 / 0.0 | 13.6 / 15.4 / 11.8 |
| **Cosine[bert-base]** | 61.9 / 27.0 | 57.9 / 16.4 | 32.1 | 57.6 | 85.0 / 63.9 / 58.6 / 41.8 | 48.9 / 8.4 | 46.6 / 57.2 / 36.0 |
| **Cosine[mpnet-base]** | 68.9 / 31.9 | 63.8 / 11.1 | 31.5 | 62.1 | 81.4 / 57.9 / 54.4 / 42.7 | 48.8 / 10.7 | 47.1 / 59.4 / 34.8 |
| **+ Coverage** | 65.0 / 33.7 | 55.1 / 17.4 | 31.1 | 57.9 | 77.5 / 67.2 / 64.0 / 48.5 | 46.2 / 11.5 | 47.9 / 55.5 / 40.4 |
| **- reorder** | 65.2 / 36.0 | 55.4 / 15.5 | 30.4 | 58.8 | 78.6 / 71.4 / 63.3 / 49.1 | 45.6 / 12.1 | 48.4 / 55.7 / 41.2 |
| **BM25** | 70.9 / 40.4 | 62.3 / 17.6 | 37.0 | 66.8 | 85.0 / 64.9 / 60.1 / 47.3 | 57.7 / 24.3 | 52.9 / 63.3 / 42.4 |
| **+ Coverage** | 66.5 / 37.6 | 51.3 / 17.8 | 31.6 | 58.8 | 76.8 / 57.9 / 55.4 / 41.8 | 48.3 / 17.9 | 46.8 / 55.6 / 38.1 |
| **- reorder** | 66.8 / 37.6 | 50.7 / 17.8 | 32.7 | 58.8 | 78.2 / 61.2 / 53.8 / 41.2 | 47.3 / 17.9 | 47.0 / 55.7 / 38.3 |
| **BM25[4-gram]** | 66.4 / 34.6 | 62.2 / 22.0 | 37.7 | 63.3 | 83.9 / 68.2 / 59.3 / 49.4 | 57.3 / 20.5 | 52.1 / 61.8 / 42.3 |
| **+ Coverage** | 72.6 / 45.0 | 60.7 / 29.4 | 35.4 | 62.9 | 81.4 / 69.1 / 61.8 / 47.6 | 60.3 / 34.4 | 55.0 / 62.2 / 47.9 |
| **- reorder** | 72.2 / 45.5 | 59.4 / 30.3 | 35.7 | 63.9 | 82.9 / 71.6 / 58.9 / 49.7 | 60.6 / 35.6 | 55.5 / 62.4 / 48.6 |
| **BM25[4-depst]** | 63.5 / 35.8 | 60.6 / 20.1 | 36.3 | 64.3 | 81.8 / 68.7 / 61.0 / 52.7 | 51.7 / 10.1 | 50.5 / 59.7 / 41.4 |
| **+ Coverage** | 72.1 / 45.1 | 60.2 / 26.2 | 33.8 | 63.8 | 84.6 / 68.5 / 61.7 / 49.7 | 58.5 / 35.0 | 54.9 / 62.2 / 47.7 |
| **- reorder** | 74.0 / 45.7 | 60.7 / 28.3 | 34.0 | 65.0 | 82.1 / 71.6 / 59.9 / 48.2 | 59.4 / 34.5 | 55.3 / 62.5 / 48.0 |
| **BSR[bert-base]** | 71.8 / 44.6 | 63.4 / 19.9 | 38.9 | 69.6 | 87.1 / 73.9 / 63.3 / 44.2 | 61.9 / 24.7 | 55.3 / 65.5 / 45.1 |
| **+ Coverage** | 76.6 / 51.9 | 66.0 / 27.5 | 39.9 | 72.7 | 88.2 / 76.3 / 62.2 / 53.0 | 69.0 / 50.5 | 61.2 / 68.7 / 53.6 |
| **BSF1[deberta-base]** | 69.7 / 33.3 | 65.6 / 18.1 | 39.3 | 67.0 | 85.0 / 69.6 / 60.4 / 47.0 | 66.9 / 23.2 | 53.8 / 65.6 / 41.9 |
| **BSP[deberta-base]** | 60.6 / 24.5 | 64.7 / 15.0 | 38.0 | 53.7 | 85.0 / 67.2 / 57.4 / 45.8 | 52.0 / 7.8 | 47.6 / 59.0 / 36.3 |
| **BSR[deberta-base]** | 70.6 / 38.3 | 64.7 / 15.1 | 39.7 | 68.2 | 85.0 / 72.2 / 60.4 / 44.8 | 66.9 / 28.1 | 54.5 / 65.9 / 43.2 |
| **+ IDF** | 71.2 / 39.3 | 64.1 / 18.7 | 38.0 | 68.1 | 83.6 / 71.6 / 59.4 / 44.5 | 63.1 / 31.8 | 54.5 / 64.7 / 44.2 |
| **+ Coverage** | 77.3 / 50.4 | 66.9 / 25.9 | 40.0 | 71.7 | 87.1 / 75.5 / 63.6 / 50.3 | 69.3 / 53.7 | 61.0 / 68.7 / 53.2 |
| **+ IDF** | 78.0 / 51.9 | 66.5 / 28.2 | 41.1 | 70.7 | 87.1 / 77.3 / 62.2 / 53.6 | nan / 53.5 | 60.9 / 68.7 / 54.5 |
| **- reorder** | 78.0 / 52.9 | 66.7 / 29.2 | 39.5 | 72.2 | 87.5 / 78.2 / 62.9 / 55.5 | 68.6 / 54.1 | 62.1 / 68.7 / 55.5 |
| **BSR[deberta-large]** | 72.1 / 38.3 | 63.5 / 16.4 | 39.4 | 69.2 | 85.4 / 72.0 / 62.6 / 45.5 | 66.3 / 27.5 | 54.8 / 66.0 / 43.7 |
| **+ IDF** | 72.2 / 40.7 | 64.9 / 16.2 | 39.5 | 69.2 | 84.3 / 73.1 / 61.7 / 44.5 | 63.9 / 33.2 | 55.3 / 65.7 / 44.9 |
| **+ Coverage** | 78.2 / 50.6 | 67.3 / 27.1 | 39.7 | 73.4 | 86.8 / 76.1 / 64.8 / 54.5 | 70.7 / 50.1 | 61.6 / 69.3 / 53.9 |
| **+ IDF** | 78.0 / 52.0 | 65.5 / 27.6 | 38.4 | 72.5 | 87.1 / 77.5 / 64.9 / 51.5 | nan / 49.3 | 60.4 / 68.3 / 53.8 |
| **- reorder** | 79.0 / 52.4 | 68.2 / 29.8 | 39.3 | 73.1 | 87.9 / 76.0 / 64.2 / 57.3 | 70.2 / 49.9 | 62.3 / 69.6 / 54.9 |

Table 18: 8-shot ICL results with StarCoder for all ablations of learning-free methods on semantic parsing datasets and splits.

| Dataset
Split
Selector | ATIS
IID / Tpl. | Overnight
IID / Tpl. | Break
IID | MTOP
IID | GeoQuery
IID / Tpl. / TMCD / Len. | SMCalFlow-CS
8_S / 32_C | AVERAGE
All / IID / Comp. |
|---|---|---|---|---|---|---|---|
| **Random** | 19.8 / 14.1 | 8.4 / 2.5 | 14.5 | 4.8 | 26.8 / 27.2 / 17.8 / 3.3 | 17.1 / 0.0 | 13.0 / 15.2 / 10.8 |
| **Cosine[bert-base]** | 63.3 / 29.3 | 49.5 / 16.4 | 33.4 | 54.8 | 77.1 / 57.3 / 54.7 / 38.2 | 49.7 / 9.2 | 44.4 / 54.6 / 34.2 |
| **Cosine[mpnet-base]** | 64.2 / 32.8 | 58.1 / 12.0 | 30.2 | 54.6 | 76.1 / 56.5 / 55.1 / 39.1 | 48.8 / 11.2 | 44.9 / 55.3 / 34.4 |
| **BM25** | 71.6 / 39.5 | 52.5 / 16.9 | 36.3 | 60.5 | 78.9 / 63.3 / 58.1 / 41.2 | 59.8 / 24.6 | 50.3 / 59.9 / 40.6 |
| **+ Coverage** | 72.1 / 48.7 | 47.9 / 25.4 | 35.5 | 61.3 | 77.1 / 65.2 / 57.2 / 49.7 | 61.5 / 38.5 | 53.3 / 59.2 / 47.4 |
| **BSR[bert-base]** | 72.3 / 40.0 | 55.3 / 19.2 | 37.9 | 63.7 | 85.4 / 68.0 / 59.1 / 39.1 | 63.1 / 26.8 | 52.5 / 62.9 / 42.0 |
| **+ Coverage** | 78.2 / 51.5 | 58.4 / 25.9 | 38.8 | 65.1 | 83.2 / 71.3 / 62.0 / 51.5 | 66.5 / 51.4 | 58.7 / 65.0 / 52.3 |
| **BSR[deberta-large]** | 73.2 / 39.3 | 55.6 / 17.6 | 38.6 | 63.2 | 82.1 / 69.7 / 56.3 / 40.3 | 66.5 / 29.7 | 52.7 / 63.2 / 42.1 |
| **+ Coverage** | 78.6 / 53.3 | 58.2 / 25.9 | 39.5 | 66.9 | 83.9 / 74.1 / 61.3 / 52.4 | 71.6 / 55.1 | 60.1 / 66.5 / 53.7 |

Table 19: 8-shot ICL results with GPT-3.5-Turbo for all ablations of learning-free methods on semantic parsing datasets and splits.

| Dataset
Split
Selector | ATIS
IID / Tpl. | Overnight
IID / Tpl. | Break
IID | MTOP
IID | GeoQuery
IID / Tpl. / TMCD / Len. | SMCalFlow-CS
8_S / 32_C | AVERAGE
All / IID / Comp. |
|---|---|---|---|---|---|---|---|
| **Random** | 16.9 / 11.1 | 5.1 / 2.8 | 13.0 | 6.8 | 22.1 / 24.9 / 15.7 / 5.5 | 20.1 / 0.0 | 12.0 / 14.0 / 10.0 |
| **Cosine[mpnet-base]** | 64.6 / 32.5 | 60.7 / 6.2 | 30.6 | 59.7 | 81.1 / 56.1 / 50.3 / 42.1 | 46.7 / 8.0 | 44.9 / 57.2 / 32.5 |
| **+ Coverage** | 59.0 / 31.6 | 52.7 / 12.0 | 27.8 | 53.3 | 76.1 / 58.6 / 56.3 / 46.7 | 46.1 / 8.1 | 44.0 / 52.5 / 35.5 |
| **- reorder** | 59.4 / 34.7 | 53.1 / 11.1 | 27.8 | 54.7 | 75.0 / 59.3 / 54.5 / 47.0 | 45.3 / 8.7 | 44.2 / 52.6 / 35.9 |
| **BM25** | 67.1 / 35.1 | 58.3 / 11.8 | 34.5 | 64.4 | 81.4 / 61.6 / 56.8 / 47.9 | 56.2 / 21.0 | 49.7 / 60.3 / 39.0 |
| **+ Coverage** | 61.3 / 33.9 | 48.4 / 10.9 | 29.6 | 58.5 | 73.9 / 50.4 / 56.2 / 41.2 | 48.3 / 12.4 | 43.7 / 53.3 / 34.1 |
| **- reorder** | 62.4 / 32.8 | 48.1 / 10.7 | 29.5 | 58.2 | 75.4 / 51.7 / 51.2 / 37.9 | 46.2 / 12.8 | 43.1 / 53.3 / 32.9 |
| **BM25[4-gram]** | 62.8 / 31.6 | 57.2 / 15.7 | 36.5 | 60.6 | 81.4 / 63.4 / 57.1 / 43.3 | 57.3 / 17.9 | 48.7 / 59.3 / 38.2 |
| **+ Coverage** | 68.0 / 40.2 | 56.2 / 18.0 | 33.7 | 61.3 | 80.0 / 61.9 / 60.3 / 46.4 | 57.3 / 28.1 | 50.9 / 59.4 / 42.5 |
| **- reorder** | 68.6 / 41.6 | 54.9 / 19.0 | 35.1 | 60.9 | 80.0 / 64.4 / 57.9 / 47.3 | 56.8 / 28.1 | 51.2 / 59.4 / 43.0 |
| **BM25[4-depst]** | 60.7 / 31.7 | 57.4 / 14.4 | 35.0 | 62.7 | 82.1 / 61.5 / 57.9 / 48.2 | 50.5 / 9.5 | 47.6 / 58.1 / 37.2 |
| **+ Coverage** | 69.8 / 40.0 | 55.3 / 15.8 | 32.9 | 63.0 | 81.1 / 61.7 / 56.3 / 43.6 | 55.6 / 32.0 | 50.6 / 59.6 / 41.6 |
| **- reorder** | 70.4 / 39.9 | 56.6 / 16.5 | 33.6 | 63.4 | 82.1 / 64.4 / 57.5 / 49.4 | 56.8 / 33.5 | 52.0 / 60.5 / 43.5 |
| **BSF1[deberta-base]** | 67.3 / 28.9 | 61.9 / 10.2 | 37.9 | 65.2 | 86.4 / 68.4 / 51.9 / 43.6 | 65.9 / 22.6 | 50.9 / 64.1 / 37.6 |
| **BSP[deberta-base]** | 58.1 / 22.2 | 61.2 / 10.4 | 36.9 | 53.3 | 83.2 / 65.6 / 52.8 / 44.8 | 48.5 / 6.2 | 45.3 / 56.9 / 33.7 |
| **BSR[deberta-base]** | 67.2 / 34.7 | 61.7 / 8.8 | 39.5 | 67.4 | 83.9 / 67.1 / 61.6 / 43.3 | 61.8 / 24.9 | 51.8 / 63.6 / 40.1 |
| **+ IDF** | 69.0 / 37.4 | 61.5 / 9.7 | 36.1 | 67.7 | 82.5 / 66.7 / 58.9 / 42.4 | 60.6 / 28.8 | 51.8 / 62.9 / 40.6 |
| **+ Coverage** | 75.7 / 47.3 | 63.0 / 18.1 | 38.4 | 70.1 | 83.2 / 71.8 / 61.8 / 49.4 | 66.6 / 46.2 | 57.6 / 66.2 / 49.1 |
| **+ IDF** | 76.0 / 48.1 | 63.5 / 20.8 | 37.5 | 69.0 | 84.3 / 71.9 / 58.2 / 46.4 | nan / 48.3 | 56.7 / 66.1 / 48.9 |
| **- reorder** | 75.2 / 48.7 | 62.5 / 18.8 | 37.7 | 69.8 | 84.6 / 70.3 / 64.6 / 47.3 | 67.8 / 49.6 | 58.1 / 66.3 / 49.9 |
| **BSR[deberta-large]** | 69.4 / 36.3 | 61.0 / 10.9 | 38.3 | 68.4 | 83.9 / 67.4 / 61.4 / 46.7 | 62.4 / 24.7 | 52.6 / 63.9 / 41.2 |
| **+ IDF** | 70.1 / 37.4 | 62.2 / 10.2 | 38.9 | 68.8 | 84.3 / 68.4 / 61.3 / 45.2 | 62.4 / 30.9 | 53.3 / 64.4 / 42.2 |
| **+ Coverage** | 76.7 / 46.9 | 63.9 / 19.9 | 40.2 | 71.5 | 88.6 / 74.9 / 64.1 / 50.3 | 68.7 / 47.8 | 59.5 / 68.3 / 50.7 |
| **+ IDF** | 77.8 / 46.9 | 64.0 / 19.7 | 38.5 | 70.5 | 85.0 / 73.7 / 64.9 / 53.3 | nan / 50.2 | 58.6 / 67.2 / 51.5 |
| **- reorder** | 76.6 / 48.0 | 63.2 / 18.7 | 39.0 | 70.1 | 85.0 / 74.4 / 63.3 / 55.5 | 67.7 / 48.1 | 59.1 / 66.9 / 51.3 |

Table 20: 8-shot ICL results with Cushman for all ablations of learning-free methods on semantic parsing datasets and splits.

| Dataset
Split
Selector | ATIS
IID / Tpl. | Overnight
IID / Tpl. | Break
IID | MTOP
IID | GeoQuery
IID / Tpl. / TMCD / Len. | SMCalFlow-CS
8_S / 32_C | AVERAGE
All / IID / Comp. |
|---|---|---|---|---|---|---|---|
| **Random** | 27.3 / 14.3 | 16.1 / 6.5 | 26.7 | 9.3 | 36.4 / 28.3 / 25.7 / 19.1 | 31.9 / 7.4 | 20.7 / 24.6 / 16.9 |
| **Cosine[mpnet-base]** | 74.1 / 39.5 | 67.7 / 17.1 | 41.0 | 69.1 | 86.8 / 68.4 / 61.5 / 48.5 | 54.7 / 11.9 | 53.4 / 65.6 / 41.1 |
| **+ Coverage** | 71.3 / 44.8 | 62.8 / 31.5 | 37.9 | 68.2 | 85.4 / 76.5 / 71.9 / 60.0 | 54.7 / 26.1 | 57.6 / 63.4 / 51.8 |
| **- reorder** | 71.7 / 45.1 | 62.6 / 29.8 | 36.4 | 67.5 | 85.0 / 78.3 / 72.5 / 59.4 | 53.8 / 25.9 | 57.3 / 62.8 / 51.8 |
| **BM25** | 76.9 / 46.7 | 66.5 / 30.1 | 45.2 | 72.3 | 87.1 / 77.2 / 71.2 / 62.1 | 65.4 / 29.4 | 60.9 / 68.9 / 52.8 |
| **+ Coverage** | 74.4 / 46.9 | 61.2 / 29.9 | 38.9 | 66.7 | 83.6 / 70.2 / 67.4 / 52.4 | 55.7 / 29.9 | 56.4 / 63.4 / 49.5 |
| **- reorder** | 74.2 / 46.7 | 60.1 / 29.2 | 39.6 | 65.3 | 83.9 / 69.3 / 66.6 / 54.2 | 55.7 / 29.0 | 56.2 / 63.1 / 49.2 |
| **BM25[4-gram]** | 73.0 / 39.9 | 65.7 / 33.8 | 43.7 | 70.4 | 86.8 / 78.6 / 70.6 / 60.6 | 63.1 / 22.8 | 59.1 / 67.1 / 51.0 |
| **+ Coverage** | 80.3 / 55.2 | 65.4 / 43.0 | 42.2 | 70.4 | 86.4 / 81.2 / 73.4 / 61.5 | 68.6 / 46.8 | 64.5 / 68.9 / 60.2 |
| **- reorder** | 79.7 / 54.9 | 65.9 / 42.4 | 41.8 | 71.1 | 87.9 / 80.1 / 73.1 / 61.8 | 67.7 / 44.2 | 64.2 / 69.0 / 59.4 |
| **BM25[4-depst]** | 70.7 / 41.1 | 65.5 / 31.2 | 42.6 | 70.4 | 85.4 / 79.9 / 69.0 / 67.3 | 58.2 / 11.9 | 57.8 / 65.5 / 50.0 |
| **+ Coverage** | 80.0 / 56.4 | 64.1 / 40.3 | 40.9 | 70.3 | 88.9 / 79.9 / 73.3 / 68.2 | 67.2 / 49.0 | 64.9 / 68.6 / 61.2 |
| **- reorder** | 80.5 / 56.4 | 64.7 / 41.0 | 39.7 | 69.4 | 87.9 / 80.2 / 73.4 / 66.4 | 67.7 / 49.3 | 64.7 / 68.3 / 61.1 |
| **BSF1[deberta-base]** | 75.4 / 39.2 | 66.9 / 28.2 | 45.2 | 73.0 | 86.1 / 78.6 / 69.6 / 52.7 | 72.1 / 27.1 | 59.5 / 69.8 / 49.2 |
| **BSP[deberta-base]** | 68.5 / 29.3 | 67.6 / 26.2 | 43.7 | 60.2 | 86.1 / 73.4 / 63.0 / 53.9 | 55.0 / 12.2 | 53.3 / 63.5 / 43.0 |
| **BSR[deberta-base]** | 75.7 / 47.1 | 67.4 / 22.2 | 45.6 | 75.7 | 88.6 / 79.7 / 69.9 / 52.4 | 72.1 / 32.0 | 60.7 / 70.8 / 50.6 |
| **+ IDF** | 76.7 / 48.9 | 67.1 / 24.6 | 44.1 | 76.3 | 87.5 / 81.0 / 69.5 / 54.5 | 70.8 / 39.5 | 61.7 / 70.4 / 53.0 |
| **+ Coverage** | 83.0 / 61.7 | 70.4 / 41.7 | 45.1 | 78.7 | 91.1 / 85.4 / 76.8 / 67.9 | 76.0 / 60.8 | 69.9 / 74.0 / 65.7 |
| **+ IDF** | 83.0 / 61.7 | 70.5 / 41.9 | 45.6 | 78.4 | 90.7 / 85.6 / 76.9 / 67.6 | nan / 62.0 | 69.4 / 73.6 / 65.9 |
| **- reorder** | 83.3 / 63.5 | 69.5 / 42.8 | 45.1 | 79.1 | 91.8 / 85.9 / 78.5 / 66.1 | 76.1 / 61.1 | 70.2 / 74.2 / 66.3 |
| **BSF1[deberta-large]** | 77.4 / 44.3 | 68.9 / 29.6 | 45.9 | 73.8 | 88.2 / 78.1 / 68.1 / 51.2 | 72.1 / 29.6 | 60.6 / 71.0 / 50.1 |
| **BSP[deberta-large]** | 70.1 / 32.5 | 69.0 / 22.9 | 44.7 | 58.9 | 87.1 / 75.6 / 61.9 / 50.6 | 63.0 / 15.7 | 54.3 / 65.5 / 43.2 |
| **BSR[deberta-large]** | 79.3 / 48.1 | 68.1 / 22.4 | 44.0 | 76.8 | 88.2 / 78.8 / 71.6 / 53.0 | 72.5 / 31.5 | 61.2 / 71.5 / 50.9 |
| **+ IDF** | 79.8 / 50.4 | 68.7 / 21.3 | 44.9 | 75.5 | 88.9 / 82.0 / 71.7 / 54.2 | 70.5 / 40.9 | 62.4 / 71.4 / 53.4 |
| **+ Coverage** | 84.7 / 62.4 | 69.5 / 41.4 | 46.0 | 79.6 | 91.1 / 86.6 / 76.0 / 69.4 | 75.7 / 61.2 | 70.3 / 74.4 / 66.2 |
| **+ IDF** | 83.8 / 62.6 | 69.4 / 39.8 | 46.5 | 78.7 | 91.1 / 86.6 / 78.6 / 68.8 | nan / 59.0 | 69.5 / 73.9 / 65.9 |
| **- reorder** | 84.0 / 64.6 | 69.6 / 39.6 | 46.4 | 79.0 | 90.0 / 86.3 / 78.0 / 70.6 | 77.5 / 60.2 | 70.5 / 74.4 / 66.5 |

Table 21: 8-shot ICL results with Codex for all ablations of learning-free methods on semantic parsing datasets and splits.