# OpenReview forum: "Coverage-based Example Selection for In-Context Learning"
_EMNLP/2023/Conference — EMNLP 2023 Findings_

### Official Review · Reviewer_cBG5 · 2023-08-01

**Soundness:** 3

**Excitement:**

3: Ambivalent: It has merits (e.g., it reports state-of-the-art results, the idea is nice), but there are key weaknesses (e.g., it describes incremental work), and it can significantly benefit from another round of revision. However, I won't object to accepting it if my co-reviewers champion it.

**Missing References:**

1) The idea of the paper is similar to [1]. Would like to have some discussions around different batch-mode selection strategies in another domain.


Reference
[1] Batch Active Learning at Scale

**Paper Topic And Main Contributions:**

Demonstration selection is a challenging task in-context learning. The general approach to example selection is to select top-k based on a metric. The paper proposes a set-based method that helps maximize the salient aspects to select examples for demonstration and show that their methodology improves over many tasks.

**Questions For The Authors:**

1) The paper uses all-mpnet-base as a base for cosine similarity but deberta-large-mnli for BERTScore. What is the impact on evaluations if different base embeddings are used for these two metrics? What are the justifications for utilizing these embeddings as the base models?

**Reasons To Accept:**

1) Proposal of  a set-based (batch mode) sample selection strategy

2) Adaption of BERTScore [1] for demonstration selection in in-context learning

3) Thorough evaluation of multiple methods, especially pointers to the failure scenarios (due to bias [1]) and scalability issues (all pair similarity computation)


Reference

[1] BERTSCORE: EVALUATING TEXT GENERATION WITH BERT

**Reasons To Reject:**

1) The performance of the proposed method is highly dependent on the scoring mechanism [1]. The proposed set-based method did not perform well using the setup in the paper for cosine and bm25 metric.

2) The paper discusses scalability issues but provides no experimental results (other than time complexity) on comparing time in their 8-shot setup. This would help readers understand performance gain compared to the computation resources requirement.



Reference

[1] BERTSCORE: EVALUATING TEXT GENERATION WITH BERT

**Reproducibility:**

4: Could mostly reproduce the results, but there may be some variation because of sample variance or minor variations in their interpretation of the protocol or method.

**Reviewer Confidence:**

4: Quite sure. I tried to check the important points carefully. It's unlikely, though conceivable, that I missed something that should affect my ratings.

---

> ### Author Rebuttal · Authors · 2023-08-29
>
> We thank the reviewer for their review.
>
> We’d first like to clarify that in addition to bertscore for ICL example selection and set-selection, our primary technical contribution is the idea of salient aspects of a test instance and the framework for measuring informativeness of demonstrations in terms of how well they demonstrate (or cover) those salient aspects.
>
> **Reasons to Reject**
>
> 1. You say that set-based selection did not perform well for cosine and BM25, but this is not what figure 3 shows. Set selection with BM25 improves over vanilla BM25 in almost all cases, and more so as the underlying LLM improves. Even with Cosine, our set-extension is significantly better in compositional splits where set-selection would be necessary. Compositional generalization is commonly required in real data settings, especially when limited training data is available. If a large amount of IID training data is available, you would likely be using some kind of fine-tuning instead of ICL.
> 2. In Line 320, we state that we found both BSR and Set-BSR to run in tens of milliseconds for nearly all datasets considered in this work—significantly faster than the LLM inference time itself. As an example, with Starcoder as the inference LLM, the selection time for all datasets and splits except DROP and QNLI was less than one-hundredth of the time for ICL with the selected examples. We will add additional runtime numbers to compare the selection time using the various methods and LLM ICL latency in the final paper.
>
> **Questions For The Authors**
>
> 1. We experimented with different encoders for both Cosine and BertScore and found it to have little impact on the performance compared to the difference between them. For Cosine we initially experimented with bert-base and for BertScore, deberta-base and large. The results for the latter are also included in the paper (Figure 9). This is also in line with the results of https://arxiv.org/abs/2302.05698 which additionally tried better sentence embedding models for Cosine. Finally, the specific choice of different encoders for Cosine and BertScore was to ensure a comparison where both are used in their best possible setting. For Cosine, all-mpnet-base-v2 is the best available and recommended encoder model in [SentenceBert](https://www.sbert.net/docs/pretrained_models.html#model-overview). For bertscore, deberta-large and base are the [best models](https://docs.google.com/spreadsheets/d/1RKOVpselB98Nnh_EOC4A2BYn8_201tmPODpNWu4w7xI/edit#gid=0) of their size. Nevertheless, we are happy to add additional results using the same encoder for both.
>
> **Missing References**
>
> 1. Thank you for the suggestion. We can add a discussion on such related work.

---

### Official Review · Reviewer_t4QT · 2023-08-05

**Soundness:** 3

**Excitement:**

3: Ambivalent: It has merits (e.g., it reports state-of-the-art results, the idea is nice), but there are key weaknesses (e.g., it describes incremental work), and it can significantly benefit from another round of revision. However, I won't object to accepting it if my co-reviewers champion it.

**Paper Topic And Main Contributions:**

The authors propose a method for selecting informative examples for in-context learning using BERTScore, which measures the similarity between two pieces of text, derived from the contextual embeddings of each token from BERT. The main contribution of this paper is the proposed BERTScore-based set metric to and the proof of its submodularity. The authors conduct experiments on multiple data sets with multiple LLMs.

**Questions For The Authors:**

Question A: What is the definition of salient aspect for cosine in Section 5? How to calculate c(s, z) if z is just a dimension?

Question B: Why do you use different sentence encoders for BERTScore and Cosine? If both use the same sentence encoders what would be the results?

Question C: Can one achieve similar improvement by simply switching to a stronger embedding (e.g., openai's embedding or other encoders with higher dimensions)?

**Reasons To Accept:**

- Research problem studied in this paper is interesting and important
- The exploration of using BERTScore-Recall in exemplar selection is novel
- Overall clear presentation

In sum, I believe that the research problems investigated in this paper are both interesting and important. The proposed method is also simple and does not require any training, which is a positive attribute.

**Reasons To Reject:**

- The definition of salient aspect for cosine is confusing, especially when it is applied in Equation (6).
- Comparisons to other set-coverage-enhanced methods like Set-BM25 / Set-cosine are not illustrated in Table 1

My biggest concern is the definition of “salient aspect” for cosine. In Equation (4), and the end of Section 4 (Line 272), the table seems to suggest that each embedding dimension is defined as the salient aspect for cosine similarity. If this definition remains the same in Section 5, it would undermine the capability of cosine in the calculation of set-coverage metric defined in Equation (6), where the max dimension will be used instead of the cosine similarity of the whole vector. The authors can probably clarify how salient aspect is defined for cosine in Section 5 and how Equation (6) is calculated for cosine.

I also think the authors can make Table 1 easier to read if they include results for Set-BM25 and Set-cosine into Table 1. This can provide clearer comparison to quantify the improvement of BSR and the set-coverage metric respectively.

**Reproducibility:**

4: Could mostly reproduce the results, but there may be some variation because of sample variance or minor variations in their interpretation of the protocol or method.

**Reviewer Confidence:**

3: Pretty sure, but there's a chance I missed something. Although I have a good feel for this area in general, I did not carefully check the paper's details, e.g., the math, experimental design, or novelty.

---

> ### Author Rebuttal · Authors · 2023-08-29
>
> We’d like to thank the reviewer for their review.
>
> **Reasons To Reject**
>
> - We’d first like to clarify that in addition to bertscore for ICL example selection and set-selection, our primary technical contribution is the idea of salient aspects of a test instance and the framework for measuring informativeness of demonstrations in terms of how well they demonstrate (or cover) those salient aspects.
>   Eq. 4 shows a way to interpret Cosine as a coverage measure (of the form of Eq. 3) so we can apply our set extension in Eq. 6 to it. It corresponds to the assumption that every dimension of the embedding vector space encodes a salient aspect and the product  $c(s, z) = r_x[s] *  r_z[s]$ measures the coverage of the dimension $s$ by example $z$. With this, cosine can be interpreted as the sum of coverage of every dimension. Set-cosine is obtained by plugging in the above $c(s ,z)$ into Eq. 6:
>
>   $$\operatorname{setcosine}(x, Z) = \sum_{s=1}^d \max_{z \in Z} c(s, z) = \sum_{s=1}^d \max_{z \in Z} r_x[s] * r_z[s]$$
>
>   The max inside sum might seem odd at first glance, but it is intended. Analogous to other set-metrics, set-cosine is the sum of maximum coverage every dimension (salient aspect) receives from any example $z \in Z$ and as our results show, this does improve performance of Cosine in compositional settings. Hope this clarifies everything about Cosine. We are happy to answer any follow-up questions.
>
> - Thank you, we’ll consider the suggestion for the final version. Currently, we use Table 1  to highlight a comparison between our best method (Set-BSR) and prior methods and dedicate an entire figure (Figure 3) to examining the influence of set-based metrics. We additionally include Tables 10-17 in the appendix with complete set of results for all methods.
>
> **Questions For The Authors**
>
> *Question A: What is the definition of salient aspect for cosine in Section 5? How to calculate c(s, z) if z is just a dimension?*
>
> As explained above, for cosine similarity, s is a dimension index and $c(s, z) = r_{x_{test}}[s] .  r_{z}[s]$ i.e. the scalar product of the values at that dimension for the test input and candidate $z$’s embeddings.
>
>
> *Question B: Why do you use different sentence encoders for BERTScore and Cosine? If both use the same sentence encoders what would be the results?*
>
> We experimented with different encoders for both Cosine and BertScore and found it to have little impact on the performance compared to the difference between them. For Cosine we initially experimented with bert-base and for BertScore, deberta-base and large. The results for the latter are also included in the paper (Figure 9). This is also in line with the results of https://arxiv.org/abs/2302.05698 which additionally tried better sentence embedding models for Cosine. Finally, the specific choice of different encoders for Cosine and BertScore was to ensure a comparison where both are used in their best possible setting. For Cosine, all-mpnet-base-v2 is the best available and recommended encoder model in [SentenceBert](https://www.sbert.net/docs/pretrained_models.html#model-overview). For bertscore, deberta-large and base are the [best models](https://docs.google.com/spreadsheets/d/1RKOVpselB98Nnh_EOC4A2BYn8_201tmPODpNWu4w7xI/edit#gid=0) of their size. Nevertheless, we are happy to add additional results using the same encoder for both.
>
>
> *Question C: Can one achieve similar improvement by simply switching to a stronger embedding (e.g., openai's embedding or other encoders with higher dimensions)?*
>
> While it’s true that better underlying models can lead to better performance -- our own experiments showed that for BERTScore (in e.g. Fig. 9 and Tables 12-17) deberta-large improved upon deberta-base (albeit marginally) -- for similarly powerful encoders, bertscore (BSR) would continue to outperform cosine as it is not bottelenecked by the use of a single embedding. Moreover, as shown in Fig. 3, set-extension works best for contextualized embeddings and BSR than for cosine similarity.

---

### Official Review · Reviewer_bmFi · 2023-08-05

**Soundness:** 3

**Excitement:**

4: Strong: This paper deepens the understanding of some phenomenon or lowers the barriers to an existing research direction.

**Paper Topic And Main Contributions:**

The authors have proposed a novel in-context learning method and experimented complex compositional generation tasks and with diverse LLMs.

**Reasons To Accept:**

Various results show nice improvements over LLMs in general, and the paper is nicely explained.

**Reasons To Reject:**

I vote for acceptance.

**Reproducibility:**

3: Could reproduce the results with some difficulty. The settings of parameters are underspecified or subjectively determined; the training/evaluation data are not widely available.

**Reviewer Confidence:**

4: Quite sure. I tried to check the important points carefully. It's unlikely, though conceivable, that I missed something that should affect my ratings.

---

> ### Author Rebuttal · Authors · 2023-08-29
>
> We thank the reviewer for their review and vote for acceptance! For reproducibility, we will make all our code and all the datasets and splits used in this paper available publicly upon acceptance.

---

### Official Review · Reviewer_YWop · 2023-08-05

**Soundness:** 3

**Excitement:**

3: Ambivalent: It has merits (e.g., it reports state-of-the-art results, the idea is nice), but there are key weaknesses (e.g., it describes incremental work), and it can significantly benefit from another round of revision. However, I won't object to accepting it if my co-reviewers champion it.

**Paper Topic And Main Contributions:**

In this paper, the authors propose to optimize set-level metrics for in-context learning to achieve better coverage of salient aspects. Experiments with several conventional large language models and generation tasks show that the proposed metric can result in better in-context learning performance by extending BERTScore.

**Questions For The Authors:**

Question A: I wonder if authors conduct some analysis in terms of computational efficiency in the experiments.

**Reasons To Accept:**

* Interesting idea on ICL measures in recall and set-level coverage.
* Extensive experiments with various LLMs and tasks.
* Concreate examples show how the proposed method improves from the cosine method.

**Reasons To Reject:**

* Violation of the anonymity policy
* Missing significance tests.
* Missing evaluation of efficiency.

**Reproducibility:**

3: Could reproduce the results with some difficulty. The settings of parameters are underspecified or subjectively determined; the training/evaluation data are not widely available.

**Reviewer Confidence:**

4: Quite sure. I tried to check the important points carefully. It's unlikely, though conceivable, that I missed something that should affect my ratings.

---

> ### Author Rebuttal · Authors · 2023-08-29
>
> We thank the reviewer for their review.
>
> **Reasons To Reject**
> 1. We have not violated the anonymity policy - our public manuscript was submitted before the arxiv deadline, and we did not publicize the paper or code during the anonymity period.
> 2. Missing significance tests: For the results in Table 1 with Codex as the inference LLM, we found that the difference between Set-BSR and all prior baselines, including CEIL, the strongest prior method, was significant with p-values < 0.001 according to paired permutation tests. We’ll add this and other significance test results to the final paper.
> 3. Missing evaluation of efficiency: We discuss the computational efficiency of bertscore-based selection through a time complexity analysis in lines 311-323. Further, as stated in the paper, in practice, it runs in tens of milliseconds for nearly all datasets we consider, adding a negligible overhead compared to the time for ICL inference itself while also not requiring any training, unlike for most prior work. As an example, with Starcoder as the inference LLM, the selection time for all datasets and splits except DROP and QNLI was less than one-hundredth of the time for ICL with the selected examples. We will add additional runtime numbers to compare the selection time using the various methods and LLM ICL latency in the final paper.
>
> **Questions For The Authors**
>
> Question A: answered above

---

### Meta-Review · Area_Chair_owkK · 2023-09-19

**Recommendation:** 5

**Metareview:**

In this paper, the authors propose to optimize set-level metrics for in-context learning to achieve better coverage of salient aspects. It proposed a BERTScore-based set metric and provides the proof of its submodularity. Experiments with several conventional LLMs and generation tasks show that the proposed metric can result in better in-context learning performance.

Strength:
1. The research problem studied in this paper (demonstration selection) is very important.
2. The proposed method is also simple and does not require any training
3. Extensive experiments with various LLMs and tasks.
4. Overall clear presentation with concrete examples.

Weakness:
1. Would be good to provide significance tests and evaluation of efficiency.
2. Lack of comparison to related area such as active learning [Batch Active Learning at Scale]

---

### Decision · Program_Chairs · 2023-10-07

**Decision:**

Accept-Findings

**Comment:**

In this paper, the authors propose to optimize set-level metrics for in-context learning to achieve better coverage of salient aspects. It proposed a BERTScore-based set metric and provides the proof of its submodularity. Experiments with several conventional LLMs and generation tasks show that the proposed metric can result in better in-context learning performance.

Strength:
1. The research problem studied in this paper (demonstration selection) is very important.
2. The proposed method is also simple and does not require any training
3. Extensive experiments with various LLMs and tasks.
4. Overall clear presentation with concrete examples.

Weakness:
1. Would be good to provide significance tests and evaluation of efficiency.
2. Lack of comparison to related area such as active learning [Batch Active Learning at Scale]